# Reproducibility of in vivo electrophysiological measurements in mice

International Brain Laboratory, Kush Banga[1], Julius Benson[2], Jai Bhagat[1], Dan Biderman[3], Daniel Birman[4], Niccolò Bonacchi[5], Sebastian A Bruijns[6], Kelly Buchanan[3], Robert AA Campbell[7], Matteo Carandini[1], Gaelle A Chapuis[8], Anne K Churchland[9]*, M Felicia Davatolhagh[9], Hyun Dong Lee[3], Mayo Faulkner[1], Berk Gerçek[8], Fei Hu[10], Julia Huntenburg[11], Cole Lincoln Hurwitz[3], Anup Khanal[9], Christopher Krasniak[12], Petrina Lau[1†], Christopher Langfield[3], Nancy Mackenzie[4], Guido T Meijer[11], Nathaniel J Miska[1], Zeinab Mohammadi[13], Jean-Paul Noel[2], Liam Paninski[3], Alejandro Pan-Vazquez[13], Cyrille Rossant[1], Noam Roth[4], Michael Schartner[11], Karolina Z Socha[1], Nicholas A Steinmetz[4], Karel Svoboda[14], Marsa Taheri[9], Anne E Urai[15], Shuqi Wang[16], Miles Wells[1], Steven J West[1], Matthew R Whiteway[3], Olivier Winter[11], Ilana B Witten[13], Yizi Zhang[3]

[1]University College London, London, United Kingdom; [2]New York University, New York, United States; [3]Columbia University, New York, United States; [4]Department of Neurobiology and Biophysics, University of Washington, Seattle, United States; [5]William James Center for Research, ISPA - Instituto Universitário, Lisbon, Portugal; [6]Max-Planck-Institute, Tübingen, Germany; [7]Sainsbury Wellcome Center, London, United Kingdom; [8]University of Geneva, Switzerland, Geneva, Switzerland; [9]University of California, Los Angeles, Los Angeles, United States; [10]University of California, Berkeley, Berkeley, United States; [11]Champalimaud Foundation, Lisbon, Portugal; [12]Cold Spring Harbor Laboratory, New York, United States; [13]Princeton University, Princeton, United States; [14]Allen Institute for Neural Dynamics WA, Seattle, United States; [15]Leiden University, Leiden, Netherlands; [16]School of Computer and Communication Sciences, EPFL, Lausanne, Switzerland

*For correspondence: AChurchland@mednet.ucla.edu

Present address: †Department of Psychiatry and Gerald Choa Neuroscience Institute, The Chinese University of Hong Kong, Shatin, Hong Kong

## eLife Assessment

This paper represents an **important** contribution to the field. Summarizing results from neural recording experiments in mice across ten labs, the work provides **compelling** evidence that basic electrophysiology features, single-neuron functional properties, and population-level decoding are fairly reproducible across labs with proper preprocessing. The results and suggestions regarding preprocessing and quality metrics may be of significant interest to investigators carrying out such experiments in their own labs.

**Abstract** Understanding brain function relies on the collective work of many labs generating reproducible results. However, reproducibility has not been systematically assessed within the context of electrophysiological recordings during cognitive behaviors. To address this, we formed a multi-lab collaboration using a shared, open-source behavioral task and experimental apparatus. Experimenters in 10 laboratories repeatedly targeted Neuropixels probes to the same location (spanning secondary visual areas, hippocampus, and thalamus) in mice making decisions; this

generated a total of 121 experimental replicates, a unique dataset for evaluating reproducibility of electrophysiology experiments. Despite standardizing both behavioral and electrophysiological procedures, some experimental outcomes were highly variable. A closer analysis uncovered that variability in electrode targeting hindered reproducibility, as did the limited statistical power of some routinely used electrophysiological analyses, such as single-neuron tests of modulation by individual task parameters. Reproducibility was enhanced by histological and electrophysiological quality-control criteria. Our observations suggest that data from systems neuroscience is vulnerable to a lack of reproducibility, but that across-lab standardization, including metrics we propose, can serve to mitigate this.

## Introduction

Reproducibility is a cornerstone of the scientific method: a given sequence of experimental methods should lead to comparable results if applied in different laboratories. In some areas of biological and psychological science, however, the reliable generation of reproducible results is a well-known challenge (*Crabbe et al., 1999*; *Sittig et al., 2016*; *Baker, 2016*; *Voelkl et al., 2020*; *Li et al., 2021*; *Errington et al., 2021*). In systems neuroscience at the level of single-cell-resolution recordings, evaluating reproducibility is difficult: experimental methods are sufficiently complex that replicating experiments is technically challenging, and many experimenters feel little incentive to do such experiments since negative results can be difficult to publish. Unfortunately, variability in experimental outcomes has been well-documented on a number of occasions. These include the existence and nature of 'preplay' (*Dragoi and Tonegawa, 2011*; *Silva et al., 2015*; *Ólafsdóttir et al., 2015*; *Grosmark and Buzsáki, 2016*; *Liu et al., 2019*), the persistence of place fields in the absence of visual inputs (*Hafting et al., 2005*; *Barry et al., 2012*; *Chen et al., 2016*; *Waaga et al., 2022*), and the existence of spike-timing dependent plasticity (STDP) in nematodes (*Zhang et al., 1998*; *Tsui et al., 2010*). In the latter example, variability in experimental results arose from whether the nematode being studied was pigmented or albino, an experimental feature that was not originally known to be relevant to STDP. This highlights that understanding the source of experimental variability can facilitate efforts to improve reproducibility.

For electrophysiological recordings, several efforts are currently underway to document this variability and reduce it through standardization of methods (*de Vries et al., 2020*; *Siegle et al., 2021*). These efforts are promising, in that they suggest that when approaches are standardized and results undergo quality control (QC), observations conducted within a single organization can be reassuringly reproducible. However, this leaves unanswered whether observations made in separate, individual laboratories are reproducible when they likewise use standardization and QC. Answering this question is critical since most neuroscience data is collected within small, individual laboratories rather than large-scale organizations. A high level of reproducibility of results across laboratories when procedures are carefully matched is a prerequisite to reproducibility in the more common scenario in which two investigators approach the same high-level question with slightly different experimental protocols. Therefore, establishing the extent to which observations are replicable even under carefully controlled conditions is critical to provide an upper bound on the expected level of reproducibility of findings in the literature more generally.

We have previously addressed the issue of reproducibility in the context of mouse psychophysical behavior, by training 140 mice in 7 laboratories and comparing their learning rates, speed, and accuracy in a simple binary visually driven decision task. We demonstrated that standardized protocols can lead to highly reproducible behavior (*Aguillon-Rodriguez et al., 2021*). Here, we build on those results by measuring within- and across-lab variability in the context of intra-cerebral electrophysiological recordings. We repeatedly inserted Neuropixels multi-electrode probes (*Jun et al., 2017*) targeting the same brain regions (including secondary visual areas, hippocampus, and thalamus) in mice performing an established decision-making task (*Aguillon-Rodriguez et al., 2021*). We gathered data across 10 different labs and developed a common histological and data processing pipeline to analyze the resulting large datasets. This pipeline included stringent new histological and electrophysiological quality-control criteria (the 'Recording Inclusion Guidelines for Optimizing Reproducibility' [RIGOR]) that are applicable to datasets beyond our own.

We define reproducibility as a lack of systematic across-lab differences: that is, the distribution of within-lab observations is comparable to the distribution of across-lab observations, and thus a data analyst would be unable to determine in which lab a particular observation was measured. This definition takes into account the natural variability in electrophysiological results. After applying the RIGOR QC measures, we found that features such as neuronal yield, firing rate, and LFP power were reproducible across laboratories according to this definition. However, the proportions of cells modulated and the precise degree of modulation by single decision-making variables, such as the sensory stimulus or the choice, while reproducible for many tests and brain regions, sometimes failed to reproduce in some regions (tests that considered the a neuron's full response profile were more robust). To interpret potential lab-to-lab differences in reproducibility, we developed a multi-task neural network encoding model that allows nonlinear interactions between variables. We found that within-lab random effects captured by this model were comparable to between-lab random effects. Taken together, these results suggest that electrophysiology experiments are vulnerable to a lack of reproducibility, but that standardization of procedures and QC metrics can help to mitigate this problem.

## Results

### Neuropixels recordings during decision-making target the same brain location

To quantify reproducibility across electrophysiological recordings, we set out to establish standardized procedures across the International Brain Laboratory (IBL) and to test whether this standardization led to reproducible results. Ten IBL labs collected Neuropixels recordings from one repeated site, targeting the same stereotaxic coordinates, during a standardized decision-making task in which head-fixed mice reported the perceived position of a visual grating (*Aguillon-Rodriguez et al., 2021*). The experimental pipeline was standardized across labs, including surgical methods, behavioral training, recording procedures, histology, and data processing *Figure 1a, b*, *Figure 1—figure supplement 1*; see Materials and methods for full details. Neuropixels probes were selected as the recording device for this study due to their standardized industrial production, and their 384 dual-band, low-noise recording channels providing the ability to sample many neurons in each of multiple brain regions simultaneously. In each experiment, Neuropixels 1.0 probes were inserted, targeted at 2.0 mm AP, -2.24 mm ML, 4.0 mm DV relative to bregma; 15° angle (*Figure 1c*). This site was selected because it encompasses brain regions implicated in visual decision-making, including visual area a/am (*Najafi et al., 2020*; *Harvey et al., 2012*), dentate gyrus (DG), CA1, (*Turk-Browne, 2019*), and lateral posterior (LP) and posterior (PO) thalamic nuclei (*Saalmann and Kastner, 2011*; *Roth et al., 2016*).

Once data were acquired and brains were processed, we visualized probe trajectories using the Allen Institute Common Coordinate Frame (*Figure 1d*, more methodological information is below). This allowed us to associate each neuron with a particular region (*Figure 1e*). To evaluate whether our data were of comparable quality to previously published datasets (*Steinmetz et al., 2019*; *Siegle et al., 2021*), we compared the neuron yield. The yield of an insertion, defined as the number of QC-passing units recovered per electrode site, is informative about the quality of both the raw recording and the spike sorting pipeline. We performed a comparative yield analysis of the repeated site insertions and the external electrophysiology datasets. Both external datasets were recorded from mice performing a visual task and included insertions from diverse brain regions.

Different spike sorting algorithms detect distinct units as well as varying numbers of units, and yield is sensitive to the inclusion criteria for putative neurons. We therefore re-analyzed the two comparison datasets using IBL's spike sorting pipeline (*Banga et al., 2022*). When filtered using identical QC metrics and split by brain region, we found the yield was comparable across datasets within each region (*Figure 1f*). A two-way ANOVA on the yield per insertion with dataset (IBL, Steinmetz, or Allen/Siegle) and region (Cortex, Thalamus, or Hippocampus) as categorical variables showed no significant effect of dataset origin (p=0.31), but a significant effect of brain region (p < 10^{-17}). Systematic differences between our dataset and others were likewise absent for a number of other metrics (*Figure 1—figure supplement 3*).

Finally, in addition to the quantitative assessment of data quality, a qualitative assessment was performed. 100 total insertions were randomly selected from the three datasets and assigned random IDs to blind the raters during manual assessment. Three raters were asked to look at snippets of raw

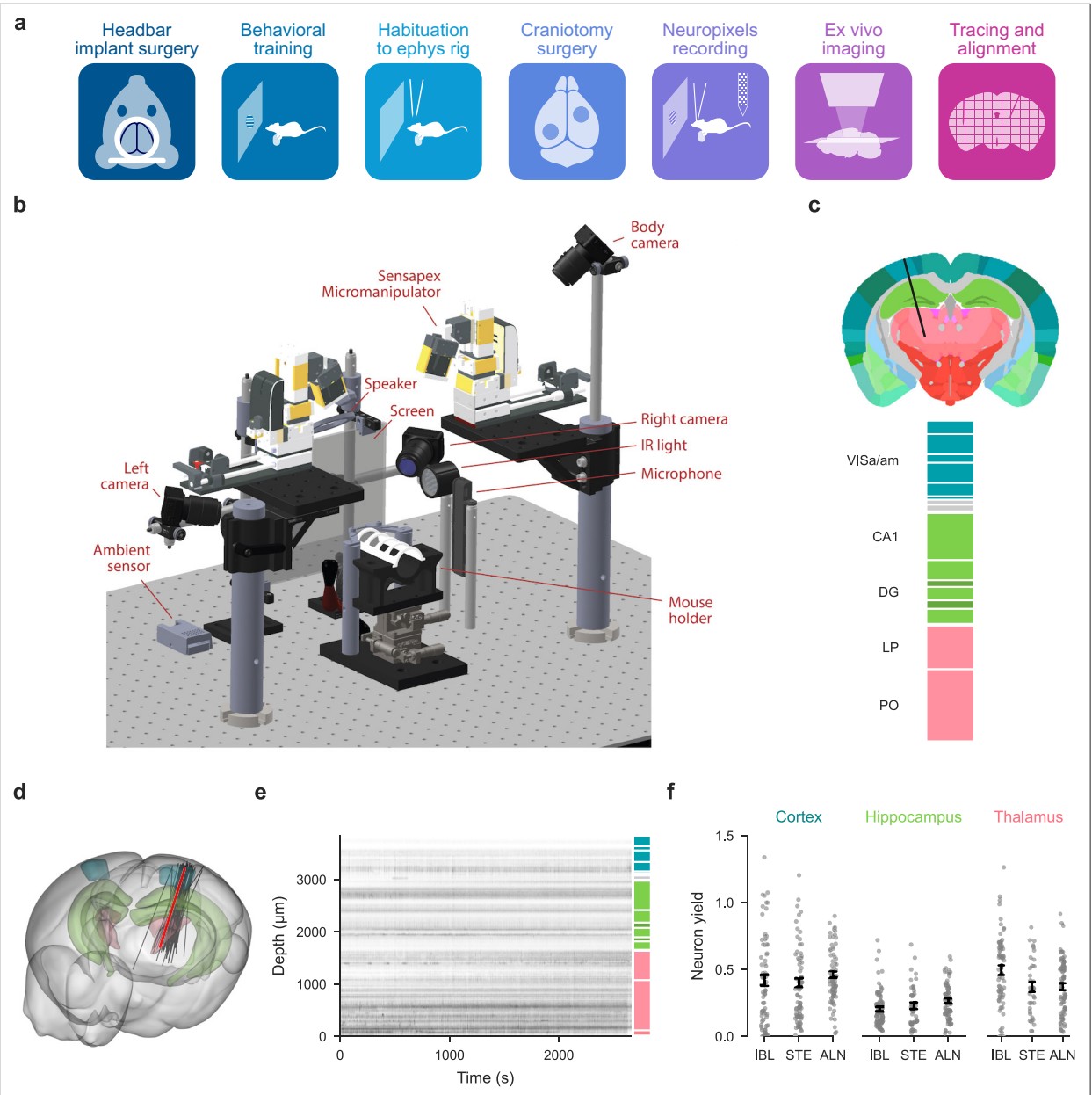

**Figure 1.** Standardized experimental pipeline and apparatus; location of the repeated site. (**a**) The pipeline for electrophysiology experiments. (**b**) Drawing of the experimental apparatus. (**c**) Location and brain regions of the repeated site. VISa: Visual Area A/AM; CA1: Hippocampal Field CA1; DG: Dentate Gyrus; LP: Lateral Posterior nucleus of the thalamus; PO: Posterior Nucleus of the Thalamus. Black lines: boundaries of sub regions within the five repeated site regions. (**d**) Gray lines: Actual repeated site trajectories shown within a 3D brain schematic. Red line: planned trajectory. (**e**) Raster plot of all measured neurons (including some that ultimately failed quality control) from one example session. (**f**) A comparison of neuron yield (neurons/ channel of the Neuropixels probe) for this dataset (IBL), *Steinmetz et al., 2019* (STE), and *Siegle et al., 2021* (ALN) in three neural structures. Bars: SEM; the center of each bar corresponds to the mean neuron yield for the corresponding study.

The online version of this article includes the following figure supplement(s) for figure 1:

**Figure supplement 1.** Detailed experimental pipeline for the Neuropixels experiment.

**Figure supplement 2.** Electrophysiology data quality examples.

**Figure supplement 3.** Detailed comparison of yield between our dataset and published reference datasets.

**Figure supplement 4.** Visual inspection of datasets by three observers blinded to data identity yielded similar metrics for all three studies.

data traces with spikes overlaid and rate the overall quality of the recording and spike detection on a scale from 1 to 10, taking into account the apparent noise levels, drift, artifacts, and missed spikes. We found no evidence for systematic quality differences across the datasets (*Figure 1—figure supplement 4*).

## Stereotaxic probe placement limits resolution of probe targeting

As a first test of experimental reproducibility, we assessed variability in Neuropixels probe placement around the planned repeated site location. Brains were perfusion-fixed, dissected, and imaged using serial section two-photon microscopy for 3D reconstruction of probe trajectories (*Figure 2a*). Whole brain auto-fluorescence data was aligned to the Allen Common Coordinate Framework (CCF; *Wang et al., 2020*) using an elastix-based pipeline (*Klein et al., 2010*) adapted for mouse brain registration (*Liu et al., 2021*). cm-DiI-labelled probe tracks were manually traced in the 3D volume (*Figure 2b*; *Figure 2—figure supplement 1*). Trajectories obtained from our stereotaxic system and traced histology were then compared to the planned trajectory. To measure probe track variability, each traced probe track was linearly interpolated to produce a straight line insertion in CCF space (*Figure 2c*).

We first compared the micro-manipulator brain surface coordinate to the planned trajectory to assess variance due to targeting strategies only (targeting variability, *Figure 2d*). Targeting variability occurs when experimenters must move probes slightly from the planned location to avoid blood vessels or irregularities. These slight movements of the probes led to sizeable deviations from the planned insertion site (*Figure 2d and g*, total mean displacement = 104 µm).

Geometrical variability, obtained by calculating the difference between planned and final identified probe position acquired from the reconstructed histology, encompasses targeting variance, anatomical differences, and errors in defining the stereotaxic coordinate system. Geometrical variability was more extensive (*Figure 2e and h*, total mean displacement = 356.0 µm). Assessing geometrical variability for all probes with permutation testing revealed a p-value of 0.19 across laboratories (*Figure 2h*), arguing that systematic lab-to-lab differences do not account for the observed variability.

To determine the extent that anatomical differences drive this geometrical variability, we regressed histology-to-planned probe insertion distance at the brain surface against estimates of animal brain size. Regression against both animal body weight and estimated brain volume from histological reconstructions showed no correlation to probe displacement ($R^2$ <0.03), suggesting differences between CCF and mouse brain sizes are not the major cause of variance. An alternative explanation is that geometrical variance in probe placement at the brain surface is driven by inaccuracies in defining the stereotaxic coordinate system, including discrepancies between skull landmarks and the underlying brain structures.

Accurate placement of probes in deeper brain regions is critically dependent on probe angle. We assessed probe angle variability by comparing the final histologically-reconstructed probe angle to the planned trajectory. We observed a consistent mean displacement from the planned angle in both medio-lateral (ML) and anterior-posterior (AP) angles (*Figure 2f and i*, total mean difference in angle from planned: 7.5 degrees). Angle differences can be explained by the different orientations and geometries of the CCF and the stereotaxic coordinate systems, which we have recently resolved in a corrected atlas (*Birman et al., 2023*). The difference in histology angle to planned probe placement was assessed with permutation testing across labs and showed a p-value of 0.45 (*Figure 2i*). This suggests that systematic lab-to-lab differences were minimal.

In conclusion, insertion variance, in particular geometrical variability, is sizeable enough to impact probe targeting to desired brain regions and poses a limit on the resolution of probe targeting. The histological reconstruction, fortunately, does provide an accurate reflection of the true probe trajectory (*Liu et al., 2021*), which is essential for interpretation of the Neuropixels recording data. We were unable to identify a prescriptive analysis to account for probe placement variance, which may reflect that the major driver of probe placement variance derives from differences in skull landmarks used for establishing the coordinate system and the underlying brain structures. Our data suggest that the resolution of probe insertion targeting on the brain surface to be approximately 360 µm (based on the mean across labs, see *Figure 2h*), which must be taken into account when planning probe insertions.

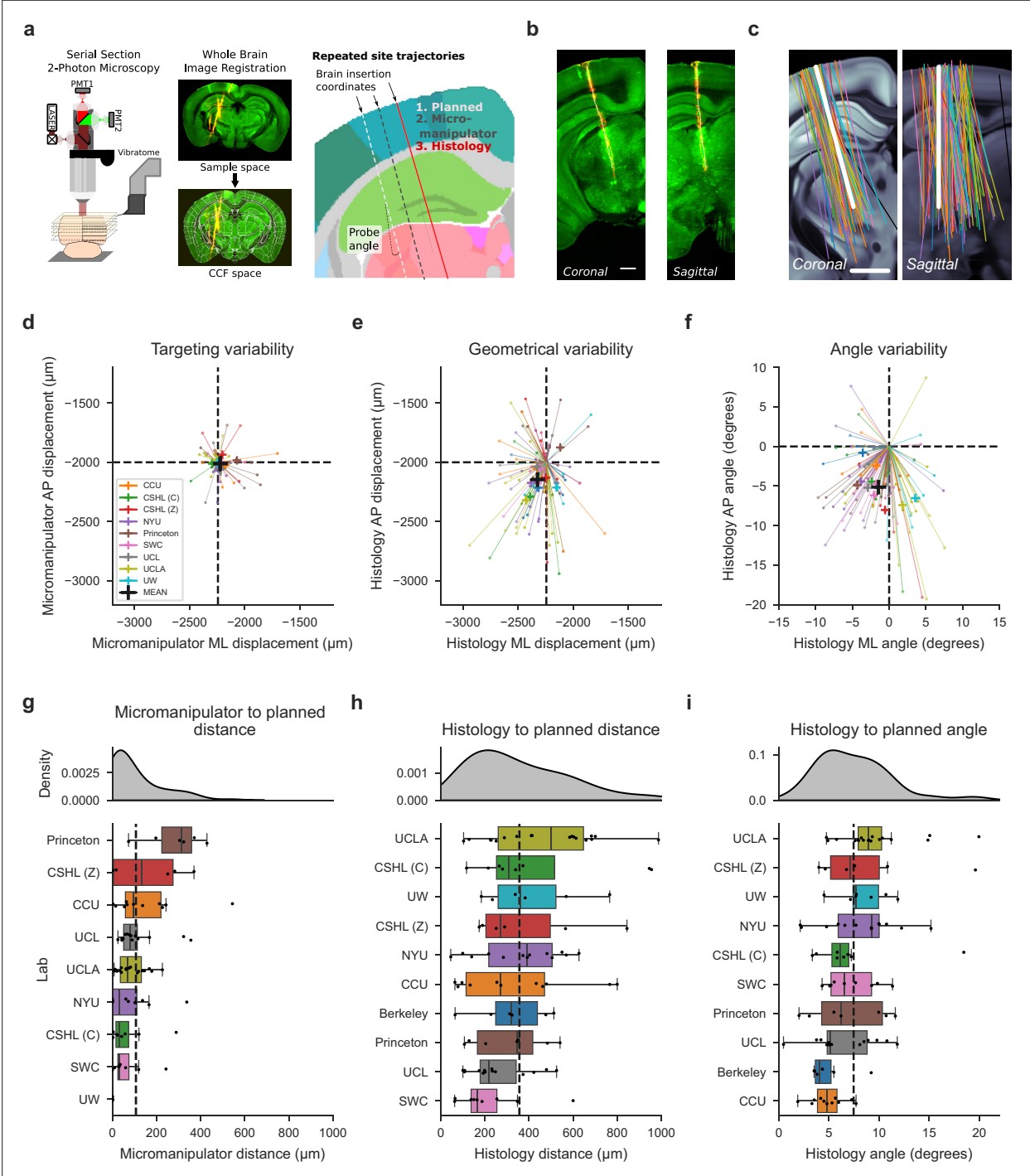

**Figure 2.** Histological reconstruction reveals resolution limit of probe targeting. (**a**) Histology pipeline for electrode probe track reconstruction and assessment. Three separate trajectories are defined per probe: planned, micro-manipulator (based on the experimenter's stereotaxic coordinates), and histology (interpolated from tracks traced in the histology data). (**b**) Example tilted slices through the histology reconstructions showing the repeated site probe track. Plots show the green auto-fluorescence data used for CCF registration and red cm-DiI signal used to mark the probe track. White dots show the projections of channel positions onto each tilted slice. Scale bar: 1 mm. (**c**) Histology probe trajectories, interpolated from traced probe tracks, plotted as 2D projections in coronal and sagittal planes, tilted along the repeated site trajectory over the allen CCF, colors: laboratory. Scale bar: 1 mm. (**d, e, f**) Scatterplots showing variability of probe placement from planned to: micro-manipulator brain surface insertion coordinate (d, targeting variability, N=88), histology brain surface insertion coordinate (**e**, geometrical variability, N=98), and histology probe angle (**f**, angle variability, N=99). Each line and point indicates the displacement from the planned geometry for each insertion in anterior-posterior (AP) and mediolateral (ML) planes, color-coded by institution. (**g, h, i**) Assessment of probe displacement by institution from planned to: micro-manipulator brain surface insertion

*Figure 2 continued on next page*

*Figure 2 continued*

coordinate (**g**, N=88), histology brain surface insertion coordinate (**h**, N=98), and histology probe angle (**i**, N=99). Kernel density estimate plots (top) are followed by boxplots (bottom) for probe displacement, ordered by descending median value. A minimum of four data points per institution led to their inclusion in the plot and subsequent analysis. Dashed vertical lines display the mean displacement across institutions, indicated in the respective scatterplot in (**d, e, f**).

The online version of this article includes the following figure supplement(s) for figure 2:

**Figure supplement 1.** Tilted slices of histology along the probe insertion for all insertions used in assessing probe placement.

## Reproducibility of electrophysiological features

Having established that targeting of Neuropixels probes to the desired target location was a source of substantial variability, we next measured the variability and reproducibility of electrophysiological features recorded by the probes. We implemented twelve exclusion criteria. Two of these were specific to our experiments: a behavior criterion where the mouse completed at least 400 trials of the behavioral task, and a session number criterion of at least three sessions per lab for analyses that directly compared across labs (permutation tests; *Figures 3d, e and 4g*, *Figure 5*). The remaining 10 criteria, which we collectively refer to as the 'RIGOR' (*Table 1*), are more general and can be applied widely to electrophysiology experiments: a yield criterion, a noise criterion, LFP power criterion, qualitative criteria for visual assessment (lack of drift, epileptiform activity, noisy channels and artifacts, see *Figure 1—figure supplement 2* for examples), and single unit metrics (refractory period violation, amplitude cutoff, and median spike amplitude). These metrics could serve as a benchmark for other studies to use when reporting results, acknowledging that a subset of the metrics will be adjusted for measurements made in different animals, regions, or behavioral contexts. The RIGOR metrics, along with additional analysis-specific constraints, determined the number of neurons, mice, and sessions included in subsequent analyses presented here (see spreadsheet).

We recorded a total of 121 sessions targeted at our planned repeated site (*Figure 3a*). Of these, 18 sessions were excluded due to incomplete data acquisition caused by a hardware failure during the experiment (10) or because we were unable to complete histology on the subject (8). Next, we applied exclusion criteria to the remaining complete datasets. We first applied the RIGOR standards described in *Table 1*. Upon manual inspection, we observed 1 recording fail due to drift, 10 recordings fail due to a high and unrecoverable count of noisy channels, 2 recordings fail due to artefacts, and 1 recording fail due to epileptiform activity. 1 recording failed our criterion for low yield, and 1 recording failed our criterion for noise level (both automated QC criteria). Next, we applied criteria specific to our behavioral experiments and found that five recordings failed our behavior criterion by not meeting the minimum of 400 trials completed. Some of our analysis also required further (stricter) inclusion criteria (see Materials and methods).

When plotting all recordings, including those that failed to meet QC criteria, one can observe that many discarded sessions were clear outliers (*Figure 3—figure supplement 1*). In subsequent figures, only recordings that passed these QC criteria were included. Overall, we analyzed data from the 82 remaining sessions recorded in 10 labs to determine the reproducibility of our electrophysiological recordings. The responses of 5312 single neurons (all passing the metrics defined in *Table 1*) are analyzed below; this total reflects an average of 105 ± 53 [mean ± std] per insertion.

We then evaluated whether electrophysiological features of these neurons, such as firing rates and LFP power, were reproducible across laboratories. In other words, is there consistent variation across laboratories in these features that is larger than expected by chance? We first visualized LFP power, a feature used by experimenters to guide the alignment of the probe position to brain regions, for all the repeated site recordings (*Figure 3b*). The DG is characterized by high-power spectral density of the LFP (*Penttonen et al., 1997*; *Bragin et al., 1995*; *Senzai and Buzsáki, 2017*), and this feature was used to guide physiology-to-histology alignment of probe positions (*Figure 3—figure supplement 2*). By plotting the LFP power of all feature aligned recordings along the length of the probe side-by-side, aligned to the boundary between the DG and thalamus, we confirmed that this band of elevated LFP power was visible in all recordings at the same depth. The variability in the extent of the band of elevated LFP power in DG was due to the fact that the DG has variable thickness and due to targeting variability, not every insertion passed through the DG at the same point in the sagittal plane (*Figure 3—figure supplement 2*).

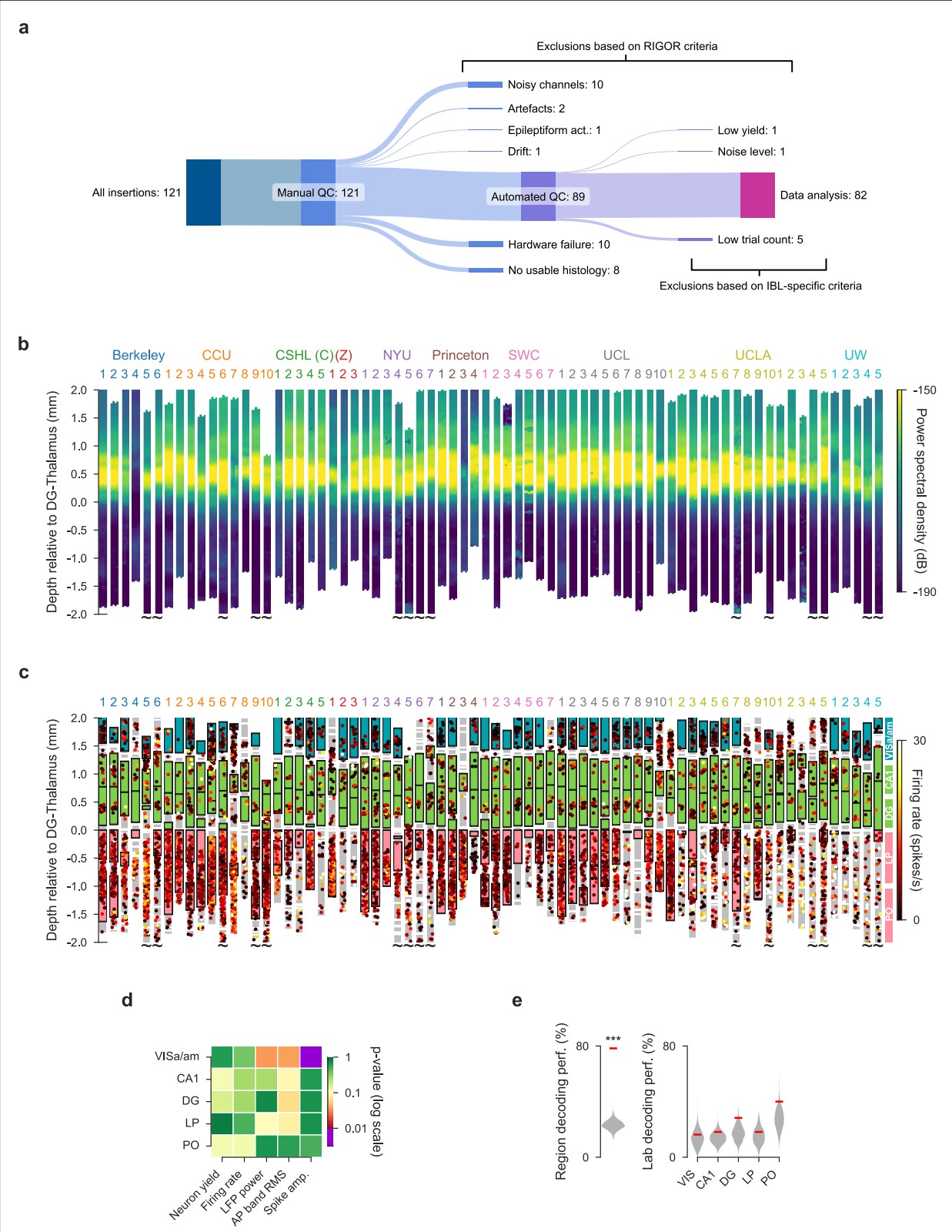

**Figure 3.** Electrophysiological features are mostly reproducible across laboratories. (**a**) Number of experimental sessions recorded; number of sessions used in analysis due to exclusion criteria. Upper branches: exclusions based on RIGOR criteria (**Table 1**); lower branches: exclusions based on experiment-specific criteria. For the rest of this figure, an additional targeting criterion was used in which an insertion had to hit two of the target regions to be included. (**b**) Power spectral density between 20 and 80 Hz of each channel of each probe insertion (vertical columns) shows reproducible

*Figure 3 continued on next page*

*Figure 3 continued*

alignment of electrophysiological features to histology. Insertions are aligned to the boundary between the dentate gyrus and thalamus. Tildes indicate that the probe continued below –2.0 mm. Location of channels reflect the data after the probes have been adjusted using electrophysiological features CSHL: Cold Spring Harbor Laboratory [(C): Churchland lab, (Z): Zador lab], NYU: New York University, SWC: Sainsbury Wellcome Centre, UCL: University College London, UCLA: University of California, Los Angeles, UW: University of Washington. (**c**) Firing rates of individual neurons according to the depth at which they were recorded. Colored blocks indicate the target brain regions of the repeated site, grey blocks indicate a brain region that was not one of the target regions. If no block is plotted, that part of the brain was not recorded by the probe because it was inserted too deep or too shallow. Each dot is a neuron, colors indicate firing rate. Probes within each institute are sorted according to their distance from the planned repeated site location. (**d**) p-values for five electrophysiological metrics, computed separately for all target regions, assessing the reproducibility of the distributions over these features across labs. p-values are plotted on a log-scale to visually emphasize values close to significance. (**e**) A Random Forest classifier could successfully decode the brain region from five electrophysiological features (neuron yield, firing rate, LFP power, AP band RMS and spike amplitude), but could not decode lab identity. The red line indicates the decoding accuracy and the grey violin plots indicate a null distribution obtained by shuffling the labels 500 times. The decoding of lab identity was performed per brain region. (* $p < 0.05$, *** $p < 0.001$).

The online version of this article includes the following figure supplement(s) for figure 3:

**Figure supplement 1.** Recordings that did not pass QC can be visually assessed as outliers.

**Figure supplement 2.** High LFP power in dentate gyrus was used to align probe locations in the brain.

**Figure supplement 3.** Bilateral recordings assess within- vs across-animal variance.

**Figure supplement 4.** Values used in the decoding analysis, per metric and per brain region.

---

The probe alignment allowed us to attribute the channels of each probe to their corresponding brain regions to investigate the reproducibility of electrophysiological features for each of the target regions of the repeated site. To visualize all the neuronal data, each neuron was plotted at the depth it was recorded overlaid with the position of the target brain region locations (*Figure 3c*). From these visualizations, it is apparent that there is recording-to-recording variability. Several trajectories missed their targets in deeper brain regions (LP, PO), as indicated by gray blocks, despite the lack of significant lab-dependent effects in targeting as reported above. These off-target trajectories tended to have both a large displacement from the target insertion coordinates and a probe angle that unfavorably drew the insertions away from thalamic nuclei (*Figure 2f*). These observations raise two questions: (1) Is the recording-to-recording variability within an animal the same or different compared to across animals? (2) Is the recording-to-recording variability lab dependent?

To answer the first question, we performed several bilateral recordings in which the same insertion was targeted in both hemispheres, as mirror images of one another. This allowed us to quantify the variability between insertions within the same animal and compare this variability to the across-animal variability (*Figure 3—figure supplement 3*). We found that whether within- or across-animal variance was larger depended on the electrophysiological feature in question (*Figure 3—figure supplement 3f*). For example, neuronal yield of VISa/am was more variable within subjects than across. By contrast, noise (assessed as action-potential band root mean square, or AP band RMS) was more variable across animals than within. Therefore, variability between metrics of bilateral recordings performed simultaneously did not bear a consistent relationship to the variability observed across subjects.

To test whether recording-to-recording variability is lab dependent, we used a permutation testing approach (Materials and methods). The tested features were neuronal yield, firing rate, spike amplitude, LFP power, and AP band RMS. These were calculated in each brain region (*Figure 3—figure supplement 4*). As was the case when analysing probe placement variability, the permutation test assesses whether the distribution over features varies significantly across labs. When correcting for multiple tests, we were concerned that systematic corrections (such as a full Bonferroni correction over all tests and regions) might be too lenient and could lead us to overlook failures (or at least threats) to reproducibility. We therefore opted to use a more stringent $\alpha$ of 0.01 when establishing significance, and refrained from applying any further corrections (this correction can be thought of as correcting only for the number of regions). The permutation test revealed a significant effect for the spike amplitude in VISa/am. Otherwise, we found that all electrophysiological features were reproducible across laboratories for all regions studied (*Figure 3d*).

The permutation testing approach evaluated the reproducibility of each electrophysiological feature separately. It could be the case, however, that some combination of these features varied systematically across laboratories. To test whether this was the case, we trained a Random Forest classifier to predict in which lab a recording was made, based on the electrophysiological markers.

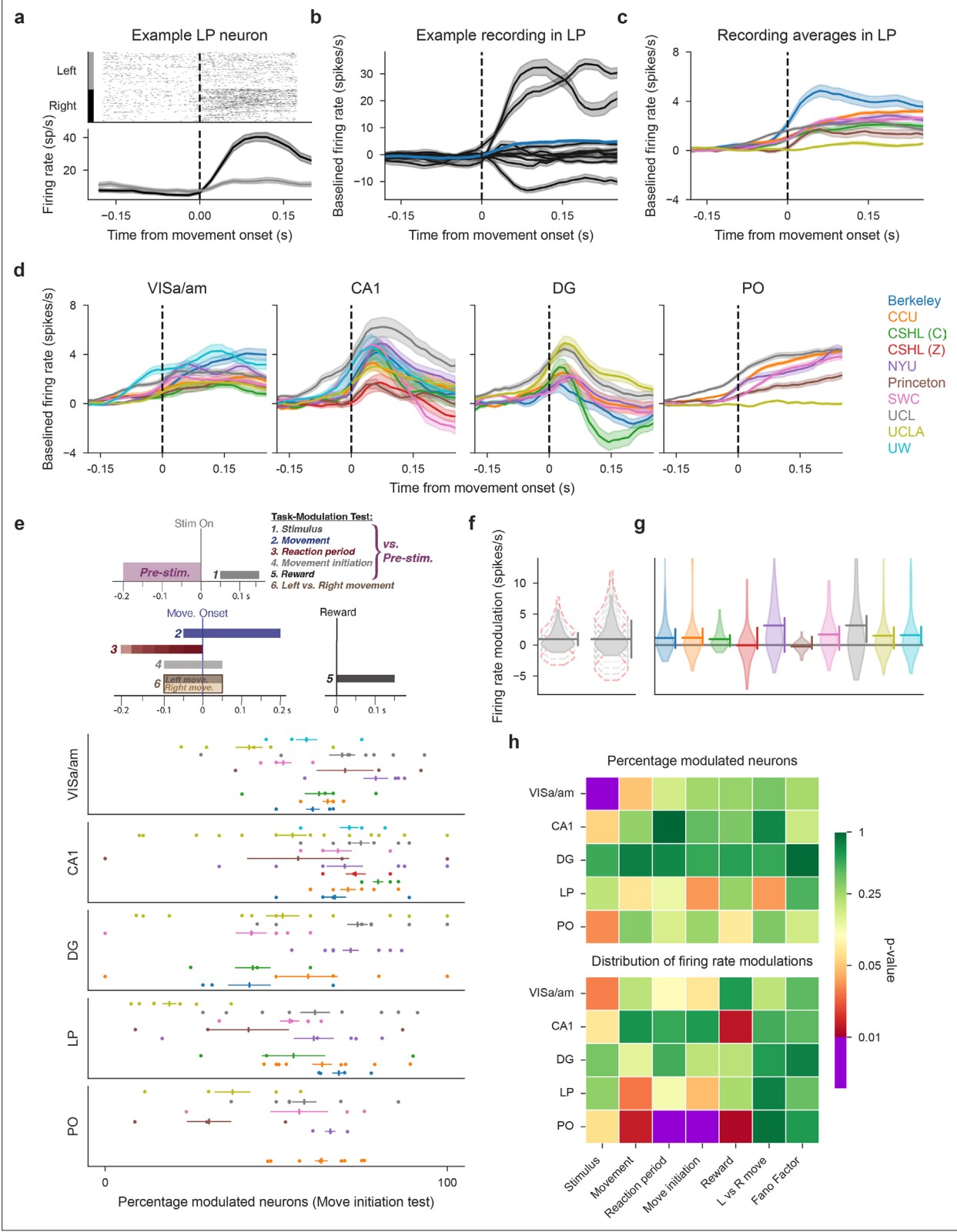

**Figure 4.** Neural activity is modulated during decision-making in five neural structures and is variable between laboratories. (**a**) Raster plot (top) and firing rate time course (bottom) of an example neuron in LP, aligned to movement onset, split for correct left and right choices. The firing rate is calculated using a causal sliding window; each time point includes a 60ms window prior to the indicated point. (**b**) Peri-event time histograms (PETHs) of all LP neurons from a single mouse, aligned to movement onset (only correct choices in response to right-side stimuli are shown). These PETHs are

*Figure 4 continued on next page*

*Figure 4 continued*

baseline-subtracted by a pre-stimulus baseline. Shaded areas show standard error of mean (and propagated error for the overall mean). The thicker line shows the average over this entire population, colored by the lab from which the recording originates. (**c,d**) Average PETHs from all neurons of each lab for LP (**c**) and the remaining four repeated site brain regions (**d**). Shaded regions indicate one standard error of the mean. (**e**) Schematic defining all six task-modulation tests (*top*) and proportion of task-modulated neurons for each mouse in each brain region for an example test (movement initiation) (*bottom*). Each row within a region correspond to a single lab (colors same as in (**d**), points are individual sessions). Horizontal lines around vertical marker: SEM around mean across lab sessions. (**f**) Schematic to describe the power analysis. Two hypothetical distributions: first, when the test is sensitive, a small shift in the distribution is enough to make the test significant (non-significant shifts shown with broken line in grey, significant shift outlined in red). By contrast, when the test is less sensitive, the vertical line is large and a corresponding large range of possible shifts is present. The possible shifts we find usually cover only a small range. (**g**) Power analysis example for modulation by the stimulus in CA1. Violin plots: distributions of firing rate modulations for each lab; horizontal line: mean across lab; vertical line at right: how much the distribution can shift up- or downwards before the test becomes significant, while holding other labs constant. (**h**) Permutation test results for task-modulated activity and the Fano Factor. Top: tests based on proportion of modulated neurons (or in the case of Fano Factor, proportion of neurons with a value <1); Bottom: tests based on the distribution of firing rate modulations (or Fano Factor values). Comparisons performed for correct trials with non-zero contrast stimuli. (Figure analyses include collected data that pass our quality metrics and have at least four good units in the specified brain region and three recordings from the specified lab, to ensure that the data from a lab can be considered representative.).

The online version of this article includes the following figure supplement(s) for figure 4:

**Figure supplement 1.** Proportion of task-modulated neurons, defined by six statistical comparisons, across mice, labs, and brain regions.

**Figure supplement 2.** Power analysis of permutation tests.

The decoding was performed separately for each brain region because of their distinct physiological signatures. A null distribution was generated by shuffling the lab labels and decoding again using the shuffled labels (500 iterations). The significance of the classifier was defined as the fraction of times the accuracy of the decoding of the shuffled labels was higher than the original labels. For validation, we first verified that the decoder could successfully identify brain region (instead of lab identity) from the electrophysiological features; the decoder was able to identify brain region with high accuracy (*Figure 3e*, left). Lab identity could not be decoded using the classifier, indicating that electrophysiology features were reproducible across laboratories for these regions (*Figure 3e*, right).

## Reproducibility of functional activity depends on brain region and analysis method

Concerns about reproducibility extend not only to electrophysiological properties but also to functional properties. To address this, we analyzed the reproducibility of neural activity driven by perceptual decisions about the spatial location of a visual grating stimulus (*Aguillon-Rodriguez et al., 2021*). In total, we compared the activity of 4400 neurons across all labs and regions. We focused on whether the targeted brain regions have comparable neural responses to stimuli, movements, and rewards. An inspection of individual neurons revealed clear modulation by, for instance, the onset of movement to report a particular choice (*Figure 4a*). Despite considerable neuron-to-neuron variability (*Figure 4b*), the temporal dynamics of session-averaged neural activity were similar across labs in some regions (see VISa/am, CA1 and DG in *Figure 4d*). Elsewhere, the session-averaged neural response was more variable (see LP and PO in *Figure 4c and d*).

Having observed that many individual neurons are modulated by task variables during decision-making, we examined the reproducibility of the proportion of modulated neurons. Within each brain region, we compared the proportion of neurons that were modulated by stimuli, movements, or rewards (*Figure 4e*). We used six different comparisons of task-related time windows (using Wilcoxon signed-rank tests and Wilcoxon rank-sum tests, *Steinmetz et al., 2019*) to identify neurons with significantly modulated firing rates to different task aspects (*Figure 4e*, top and *Figure 4—figure supplement 1* specify which time windows we compared for which test on each trial; for this, we use $\alpha = 0.05$; since leftwards versus rightwards choices are not paired we use Wilcoxon rank-sum tests to determine modulation for them). For most individual tests, the proportions of modulated neurons across sessions and across brain regions were quite variable (*Figure 4e*, bottom and *Figure 4—figure supplement 1*). We also measured the neuronal Fano Factor, which enables the comparison of the fidelity of signals across neurons and regions despite differences in firing rates (*Tolhurst et al., 1983*). Since the Fano Factor tends to be consistently low around movement time (*Churchland et al., 2010*; *Churchland*

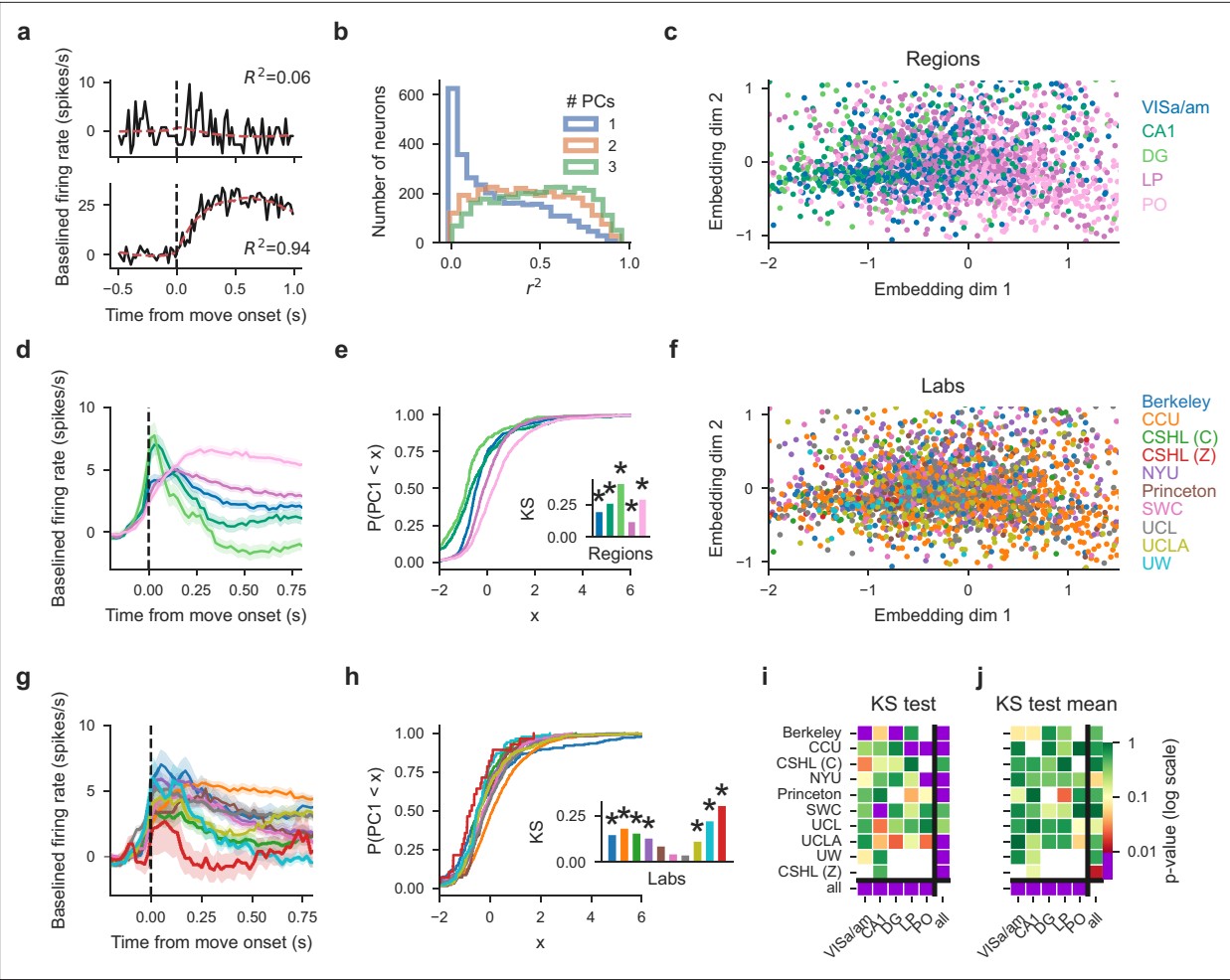

**Figure 5.** Principal component embedding of trial-averaged activity uncovers differences that are clear region-to-region and more modest lab-to-lab. (**a**) PETHs from two example cells (black, fast reaction times only) and 2-PC-based reconstruction (red). Goodness of fit $R^2$ indicated on top with an example of a poor (top) and good (bottom) fit. (**b**) Histograms of reconstruction goodness of fit across all cells based on reconstruction by 1–3 PCs. Since PETHs are well approximated with just the first 2 PCs, subsequent analyses used the first 2 PCs only. (**c**) Two-dimensional embedding of PETHs of all cells colored by region (each dot corresponds to a single cell). (**d**) Mean firing rates of all cells per region, note visible pink/green divide in line with the scatter plot. Error bands are SD across cells normalized by the square root of the number of cells in the region (**e**) Cumulative distribution of the first embedding dimension (PC1, as shown in (**c**)) per region with inset of KS statistic, measuring the maximum absolute difference between the cumulative distribution of a region's first PC values and that of the remaining cells; asterisks indicate significance at p = 0.01. (**f**) same data as in (**c**) but colored by lab. Visual inspection does not show lab clusters. (**g**) Mean activity for each lab, using cells from all regions (color conventions the same as in (**f**)). Error bands are SD across cells normalized by square root of number of cells in each lab. (**h**) same as (**e**) but grouping cells per lab. (**i**) p-values of all KS tests without sub-sampling; white squares indicate that there were too few cells for the corresponding region/lab pair. The statistic is the KS distance of the distribution of a target subset of cells' first PCs to that of the remaining cells. Columns: the region to which the test was restricted and each row is the target lab of the test. Bottom row 'all': p-values reflecting a region's KS distance from all other cells. Rightmost column 'all': p-values of testing a lab's KS distance from all other cells. Small p-values indicate that the target subset of cells can be significantly distinguished from the remaining cells. Note that all region-targeting tests are significant while lab-targeting tests are less so. (**j**) Cell-number-controlled test results; for the same tests as in i the the minimum cell number across compared classes was sampled from the others before the test was computed and p-values combined using Fisher's method across samplings. Fewer tests are significant after this control. (Figure analyses include collected data that pass our quality metrics and have at least four good units in the specified brain region and three recordings from the specified lab, to ensure that the data from a lab can be considered representative.).

The online version of this article includes the following figure supplement(s) for figure 5:

**Figure supplement 1.** Lab-grouped average PETH, CDF of the first PC and 2-PC embedding, separate per brain region.

**Table 1.** Recording Inclusion Guidelines for Optimizing Reproducibility (RIGOR).
These criteria could be applied to other electrophysiology experiments (including those using other probes) to enhance reproducibility. We provide code to apply these metrics to any dataset easily (see Code Availability). Note that whole recording metrics are most relevant to large scale (>20 channel) recordings. For those recordings, manual inspection of datasets passing these criteria will further enhance data quality.

| | | Criterion | Definition |
|---|---|---|---|
| | Computed | Yield | At least 0.1 neurons (that pass single unit criteria) per electrode channel in each brain region. |
| | | Noise level | Median action-potential band RMS (AP RMS) less than 40 µV |
| | | LFP derivative | Median of the derivative of the LFP band (20–80 Hz) less than 0.05 dB/µm (may differ for other electrodes). |
| | Visually assessed | Drift | Drift here refers to the relative movement of brain tissue and electrodes, measured as a function of depth and time. Qualitatively, we look for sharp displacements as a function of time, as observed on the raster plot; severe discontinuity will cause units to artifactually disappear/reappear. Quantitatively, the drift is the cumulative absolute electrode motion estimated during spike sorting (µm). |
| | | Epileptiform activity | Absence of epileptiform activity, which is characterized by sharp discontinuities on the raster plot (not driven by movement or noise artifacts) or strong periodic spiking spanning many channels. |
| | | Noisy channels | Absence of noisy or poor impedance channel groups (e.g. lack of visible action potential on the raw data plot). |
| Whole recording | | Artefacts | Absence of artefacts, which are characterised by a sudden discontinuity in the raw signal, spanning nearly all channels at once. |
| Single unit | Computed | Refractory period violation | Each neuron must pass a sliding refractory period metric, a false positive estimate which computes the confidence that a neuron has below 10% contamination for all possible refractory period lengths (from 0.5 to 10ms; see Materials and methods). A neuron passes if the confidence metric is greater than 90% for any possible refractory period length. |
| | | Amplitude cutoff | Each neuron must pass a metric that estimates the number of spikes missing (false negative rate) and ensures that the distribution of spike amplitudes is not cut off, without a Gaussian assumption. See Materials and methods for description of quantification and thresholds. |
| | | Median spike amplitude | Each neuron must have a median spike amplitude greater than 50 µV. |

*et al., 2011*), we calculated the average Fano Factor per neuron during 40–200ms after movement onset (for correct trials with full-contrast right-side stimuli) and quantified differences across labs.

After determining that our measurements afforded sufficient power to detect differences across labs (*Figure 4f, g*, *Figure 4—figure supplement 2*, see Materials and methods), we evaluated reproducibility of these proportions using permutation tests (*Figure 4h*). (Note that we report the reproducibility of the Fano Factor across labs with some caution due to our power analysis results in *Figure 4—figure supplement 2*). Our tests uncovered only one region and period for which there were significant differences across labs ($\alpha = 0.01$, corresponding to a Bonferroni correction for the number of regions but not the number of tests, as described above): VISa/am during the stimulus onset period (*Figure 4h*, top). In addition to examining the proportion of responsive neurons (a quantity often used to determine that a particular area subserves a particular function), we also compared the full distribution of firing rate modulations. Here, reproducibility was tenuous and failed in thalamic regions for some tests (*Figure 4h*, bottom). Taken together, these tests uncover that some traditional metrics of neuron modulation are vulnerable to a lack of reproducibility.

These failures in reproducibility were driven by outlier labs; for instance, for movement-driven activity in PO, UCLA reported an average change of 0 spikes/s, while CCU reported a large and consistent change (*Figure 4d*, right most panel, compare orange vs. yellow traces). Similarly, the differences in modulation in VISa/am were driven by one lab, in which all four mice had a higher percentage of modulated neurons than mice in other labs (*Figure 4—figure supplement 1a*, purple dots at right) *Figure 6*.

Looking closely at the individual sessions and neuronal responses, we were unable to find a simple explanation for these outliers: they were not driven by differences in mouse behavior, wheel kinematics when reporting choices, or spatial position of neurons (examined in detail in the next section). This high variability across individual neurons meant that differences that are clear in across-lab aggregate data can be unclear within a single lab. For instance, a difference between the proportion of neurons modulated by movement initiation vs. left/right choices is clear in the aggregate data (compare panels d and f in *Figure 4—figure supplement 1*), but much less clear in the data from any single lab (e.g. compare values for individual labs in panels d vs. f of *Figure 4—figure supplement 1*).

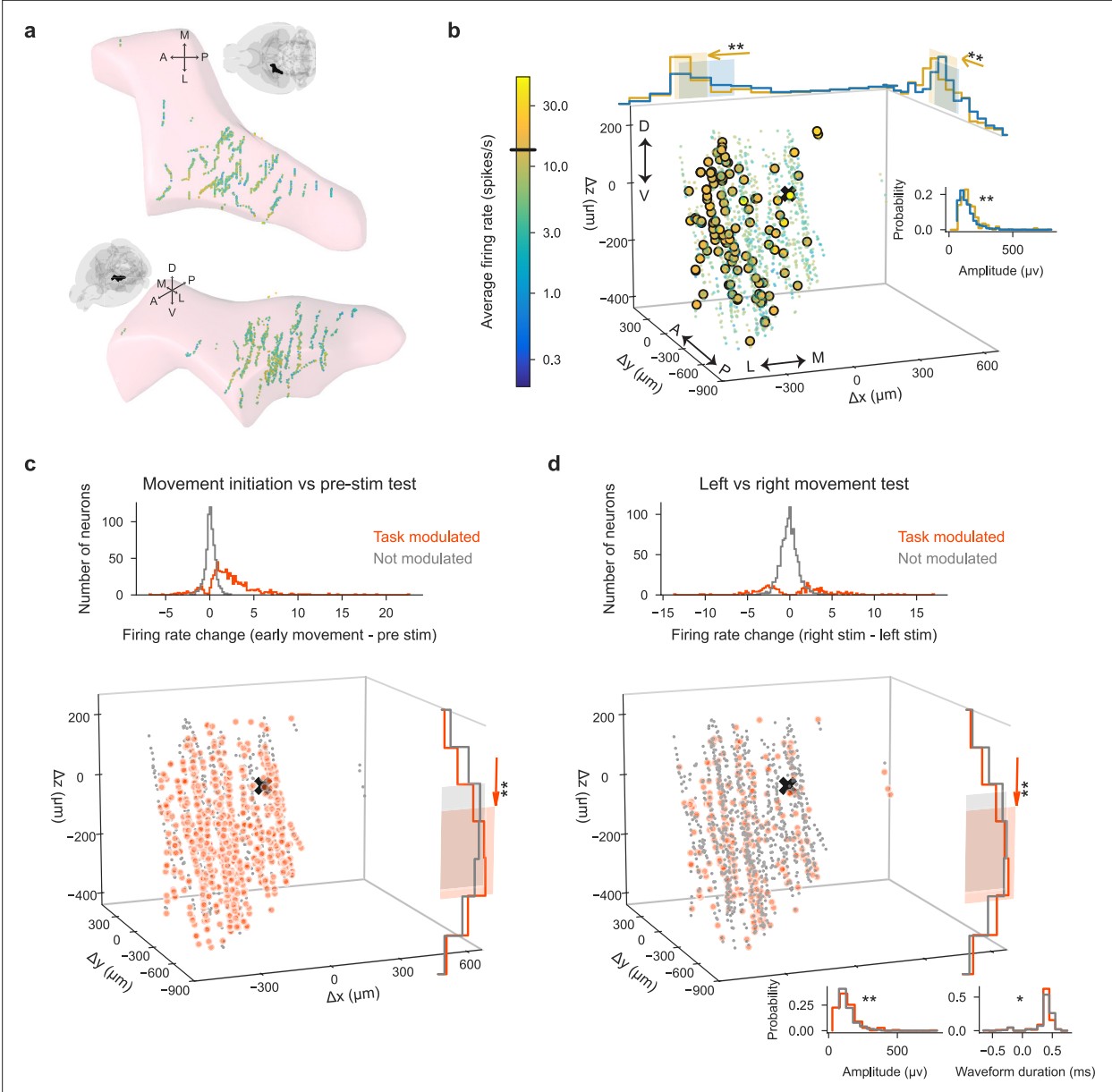

**Figure 6.** High-firing and task-modulated LP neurons have slightly different spatial positions and spike waveform features than other LP neurons, possibly contributing only marginally to variability between sessions. (**a**) Spatial positions of recorded neurons in LP. Colors: session-averaged firing rates on a log scale. To enable visualization of overlapping data points, small jitter was added to the unit locations.(**b**) Spatial positions of LP neurons plotted as distance from the planned target center of mass, indicated with the black x. Colors: session-averaged firing rates on a log scale. Larger circles: outlier neurons (above the firing rate threshold of 13.5 sp/s shown on the colorbar). In LP, 114 out of 1337 neurons were outliers. Only histograms of the spatial positions and spike waveform features that were significantly different between the outlier neurons (yellow) and the general population of neurons (blue) are shown (two-sample Kolmogorov-Smirnov test with Bonferroni correction for multiple comparisons; * and ** indicate corrected p-values of <0.05 and<0.01. Shaded areas: the area between 20th and 80th percentiles of the neurons' locations. (**c**) (Left) Histogram of firing rate changes during movement initiation (*Figure 4e*, *Figure 4—figure supplement 1d*) for task-modulated (orange) and non-modulated (gray) neurons. 697 out of 1337 LP neurons were modulated during movement initiation. (Right) Spatial positions of task-modulated and non-modulated LP neurons, with histograms of significant features (here, z position) shown. (**d**) Same as (**c**) but using the left vs. right movement test (*Figure 4e*, *Figure 4—figure supplement 1f*) to identify task-modulated units; histogram is bimodal because both left- and right-preferring neurons are included.292 out of 1337 LP neurons were modulated differently for leftward vs. rightward movements. (Figure analyses include all collected data that pass our quality metrics, regardless of the number of recordings per lab or number of repeated site brain areas that the probes pass through.).

The online version of this article includes the following figure supplement(s) for figure 6:

**Figure supplement 1.** High-firing and task-modulated VISa/am neurons.

*Figure 6 continued on next page*

This is an example of a finding that might fail to reproduce across individual labs, and underscores that single-neuron analyses are susceptible to lack of reproducibility, in part due to the high variability of single neuron responses. Very large sample sizes of neurons could serve to mitigate this problem. We anticipate that the feasibility of even larger scale recordings will make lab-to-lab comparisons easier in future experiments; multi-shank probes could be especially beneficial for cortical recordings, which tend to be the most vulnerable to low cell counts since the cortex is thin and is the most superficial structure in the brain and thus the most vulnerable to damage. Analyses that characterize responses to multiple parameters are another possible solution (See *Figure 7*).

Having discovered vulnerabilities in the reproducibility of standard single-neuron tests, we tested reproducibility using an alternative approach: comparing low-dimensional summaries of activity across labs and regions. To start, we summarized the response for each neuron by computing peri-event time histograms (PETHs, *Figure 5a*). Because temporal dynamics may depend on reaction time, we generated separate PETHs for fast ($< 0.15\,\mathrm{s}$) and slow ($> 0.15\,\mathrm{s}$) reaction times ($0.15\,\mathrm{s}$ was the mean reaction time when considering the distribution of reaction times across all sessions). We concatenated the resulting vectors to obtain a more thorough summary of each cell's average activity during decision-making. The results (below) did not depend strongly on the details of the trial-splitting; for example, splitting trials by 'left' vs 'right' behavioral choice led to similar results. We further tried a stimulus-aligned pre-movement window, which resulted in many fewer responding cells, although no substantially different test results, not shown here.

Next, we projected these high-dimensional summary vectors into a low-dimensional 'embedding' space using principal component analysis (PCA). This embedding captures the variability of the population while still allowing for easy visualization and further analysis. Specifically, we stack each cell's summary double-PETH vector (described above) into a matrix (containing the summary vectors for all

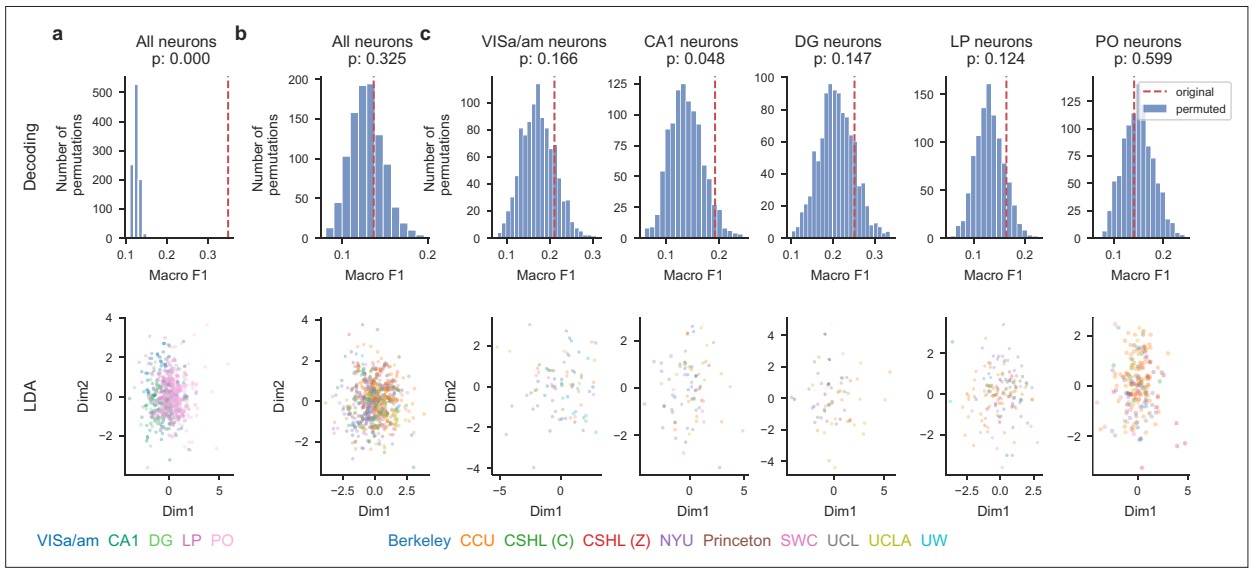

**Figure 7.** Brain region, but not lab identity is decodable from single-neuron response profiles. Results of decoding analysis for different scenarios: (**a**) brain region decoding using all neurons, (**b**) lab decoding using all neurons, and (**c**) lab decoding using neurons from specific brain regions. The histogram shows the distribution of macro-averaged F1 for perturbed null datasets. The null distribution is compared to the macro-averaged F1 of the original dataset (indicated by the dashed red line). The scatter plot shows the cross-validated two-dimensional linear discriminant analysis (LDA) projection. Neurons are randomly and evenly split into two groups: one for training the LDA model and the other for testing. The plots are generated exclusively using the test set.

cells across all sessions) and run PCA to obtain a low-rank approximation of this matrix (see Materials and methods). The accuracy of reconstructions from the top two principal components (PCs) varied across cells (*Figure 5a*); PETHs for the majority of cells could be well-reconstructed with just 2 PCs (*Figure 5b*).

This simple embedding exposes differences in brain regions (*Figure 5c*; e.g. PO and DG are segregated in the embedding space), verifying that the approach is sufficiently powerful to detect expected regional differences in response dynamics. Region-to-region differences are also visible in the region-averaged PETHs and cumulative distributions of the first PCs (*Figure 5d and e*). By contrast, differences are less obvious when coloring the same embedded activity by labs (*Figure 5f, g and h*); however some labs, such as CCU and CSHL(Z), are somewhat segregated. In comparison to regions, the activity point clouds overlap somewhat more homogeneously across most labs (*Figure 5—figure supplement 1* for scatter plots, PETHs, cumulative distributions for each region separately, colored by lab).

We quantified this observation via two tests. First, we applied a permutation test using the first 2 PCs of all cells, computing each region's 2D distance between its mean embedded activity and the mean across all remaining regions, then comparing these values to the null distribution of values obtained in an identical manner after shuffling the region labels. Second, we directly compared the distributions of the first PCs, applying the Kolmorogov-Smirnov (KS) test to determine whether the distribution of a subset of cells was different from that of all remaining cells, targeting either labs or regions. The KS test results were nearly identical to the 2D distance permutation test results, hence we report only the KS test results below.

This analysis confirmed that the neural activity in each region differs significantly (as defined above, $\alpha = 0.01$, Bonferroni correction for the number of regions but not the number of tests) from the neural activity measured in the other regions (*Figure 5i*, bottom row), demonstrating that neural dynamics differ from region-to-region, as expected and as suggested by *Figure 5c–e*. We also found that the neural activity from 7/10 labs differed significantly from the neural activity of all remaining labs when considering all neurons (*Figure 5i*, right column). When restricting this test to single regions, significant differences were observed for 6/40 lab-region pairs (*Figure 5i*, purple squares in the left columns). Note that in panels i,j of *Figure 5*, the row 'all' indicates that all cells were used when testing if a region had a different distribution of first PCs than the remaining regions and analogously the column 'all' for lab-targeted tests. We further repeated these tests after controlling for varying cell numbers, by sampling the minimum cell number across compared classes and combining p-values across these samplings using Fisher's method, which resulted in overall fewer significant differences (*Figure 5j*). Overall, these observations are in keeping with *Figure 4* and suggest, as above, that outlier responses during decision-making can be present despite careful standardization (*Figure 5i*).

## The spatial position within regions and spike waveforms of neurons explain minimal firing rate variability

In addition to lab-to-lab variability, which was low for electrophysiological features and higher for functional modulation during decision-making (*Figures 4 and 5*), we observed considerable variability between recording sessions and mice. Here, we examined whether this variability could be explained by either the within-region location of the Neuropixels probe or the single-unit spike waveform characteristics of the sampled neurons.

To investigate variability in session-averaged firing rates, we identified neurons that had firing rates different from the majority of neurons within each region (absolute deviation from the median firing rate being >15% of the firing rate range). These outlier neurons, which all turned out to be high-firing, were compared against the general population of neurons in terms of five features: spatial position (x, y, z, computed as the center-of-mass of each unit's spike template on the probe, localized to CCF coordinates in the histology pipeline) and spike waveform characteristics (amplitude, peak-to-trough duration). We observed that recordings in all areas, such as LP (*Figure 6a*), indeed spanned a wide space within those areas. Interestingly, in areas other than DG, the highest firing neurons were not entirely uniformly distributed in space. For instance, in LP, high firing neurons tended to be positioned more laterally and centered on the anterior-posterior axis (*Figure 6b*). In VISa/am and CA1, only the spatial position of neurons, but not differences in spike characteristics, contributed to differences in session-averaged firing rates (*Figure 6—figure supplements 1b and 2b*). In contrast, high-firing

neurons in LP, PO, and DG had different spike characteristics compared to other neurons in their respective regions (*Figure 6b*, *Figure 6—figure supplement 3a, c*). It does not appear that high-firing neurons in any brain region belong clearly to a specific neuronal subtype (see *Figure 6—figure supplement 5*).

To quantify variability in session-averaged firing rates of individual neurons that can be explained by spatial position or spike characteristics, we fit a linear regression model with these five features (x, y, z, spike amplitude, and duration of each neuron) as the inputs. For each brain region, features that had significant weights were mostly consistent with the results reported above: In VISa/am and CA1, spatial position explained part of the variance; in DG, the variance could not be explained by either spatial position or spike characteristics; in LP and PO, both spatial position and spike amplitudes explained some of the variance. In LP and PO, where the most amount of variability could be explained by this regression model (having higher $R^2$ values), these five features still only accounted for a total of ~5% of the firing rate variability. In VISa/am, CA1, and DG, they accounted for approximately 2%, 4%, and 2% of the variability, respectively.

Next, we examined whether neuronal spatial position and spike features contributed to variability in task-modulated activity. We found that brain regions other than DG had minor, yet significant, differences in spatial positions of task-modulated and non-modulated neurons (using the definition of at least of one of the six time-period comparisons in *Figure 4c*, *Figure 4—figure supplement 1*). For instance, LP neurons modulated according to the movement initiation test and the left versus right movement test tended to be more ventral (*Figure 6c–d*). Other brain regions had weaker spatial differences than LP (*Figure 6—figure supplements 1–3*). Spike characteristics were significantly different between task-modulated and non-modulated neurons only for one modulation test in LP (*Figure 6d*) and two tests in DG, but not in other brain regions. On the other hand, the task-aligned Fano Factors of neurons did not have any differences in spatial position except for in VISa/am, where lower Fano Factors (<1) tended to be located ventrally (*Figure 6—figure supplement 4*). Spike characteristics of neurons with lower vs. higher Fano Factors were only slightly different in VISa/am, CA1, and PO. Lastly, we trained a linear regression model to predict the 2D embedding of PETHs of each cell shown in *Figure 5c* from the x, y, z coordinates and found that spatial position contains little information ($R^2 \sim 5\%$) about the embedded PETHs of cells.

In summary, our results suggest that neither spatial position within a region nor waveform characteristics account for very much variability in neural activity. We examine other sources of variability below (see section, A multi-task neural network accurately predicts activity and quantifies sources of neural variability).

## Single neuron coefficients from a regression-based analysis are reproducible across labs

Neural encoding models provide a natural way to quantify the impact of many variables on single neuron activity, and to test whether the resulting profiles are reproducible across labs or brain regions. We first estimated single-neuron response profiles using standard linear regression models. Second, we evaluated reproducibility by decoding either lab identity or brain region from the response profiles of each neuron.

We used a Reduced-Rank Regression (RRR) encoding model to predict the single neuron activity as a function of the experimental conditions and the subject's behavior. Our approach follows a previous application of the RRR model to cortical neurons in the IBL Brainwide Map dataset (*Posani et al., 2025*). Specifically, the input variables we used for fitting the model range from cognitive (left vs. right block), sensory (stimulus side, contrast), motor (wheel velocity, whisker motion energy, lick), to decision-related (choice, outcome). The goal of the model is to relate these variables to the measured neural activity (aligned to stimulus onset, smoothed, and normalized). After fitting the RRR model to predict trial-by-trial activity, each neuron is summarized by a single 32-dimensional coefficient vector.

Using these 32-dimensional vectors, we then attempted to decode the region or lab identity associated with the neurons. We used 69 sessions from the dataset presented here. Neurons from the regions VISa, DG, PO, CA1, LP, passing the QC (RIGOR), and with sufficient goodness-of-fit (cross-validated $R^2 \geq 0.03$) were included in the decoding analysis. There were 1767 responsive neurons in total. We used the support vector classification (SVC) for the multi-class classification and the macro-averaged F1 score to assess the goodness of decoding. Additionally, we conducted a permutation

test (Materials and methods, N_permute = 1000) to determine whether the macro-averaged F1 score obtained from the data was significantly higher than chance level.

We found that the region identity was more decodable than the lab identity. The macro-averaged F1 score for the region decoding was 0.35 ($p < 0.0001$, *Figure 7a*), while for lab decoding, it was only 0.14 ($p = 0.325$, *Figure 7b*). To ensure that the unequal distribution of recorded regions did not cause us to mis-estimate the lab decoding performance, we applied the same lab decoding analysis to each region separately. The results are close to chance level for all the regions ($p = 0.166$, $0.048$, $0.147$, $0.124$ $0.599$ for VISa, CA1, DG, LP, and PO respectively, *Figure 7c*). The results align with our visual inspection using linear discriminant analysis (*Figure 7*, bottom row), where region-to-region differences are more visible than lab-to-lab differences. In summary, the identity of the brain region is significantly decodable from the single-neuron response profiles, while the identity of the laboratory is not. This speaks to the reproducibility of single neuron metrics across labs, and provides reassurance that when a neuron's full response profile is considered (instead of its selectivity for just one variable as in *Figure 4*), results are more robust.

## A multi-task neural network accurately predicts activity and quantifies sources of neural variability

As discussed above, variability in neural activity between labs or between sessions can be due to many factors. These include differences in behavior between animals, differences in probe placement between sessions, and uncontrolled differences in experimental setups between labs. Simple linear regression models or generalized linear models (GLMs) may be too inflexible to capture the nonlinear contributions that many of these variables, including lab identity and spatial positions of neurons, might make to neural activity. On the other hand, fitting a different nonlinear regression model (involving many covariates) individually to each recorded neuron would be computationally expensive and could lead to poor predictive performance due to overfitting.

To estimate a flexible nonlinear model given constraints on available data and computation time, we adapt an approach that has proven useful in the context of sensory neuroscience (*McIntosh et al., 2016*; *Batty et al., 2016*; *Cadena et al., 2019*). We use a 'multi-task' neural network (MTNN; *Figure 8a*) that takes as input a set of covariates (including the lab ID, the neuron's 3D spatial position in standardized CCF coordinates, the animal's estimated pose extracted from behavioral video monitoring, feedback times, and others; see *Table 2* for a full list). The model learns a set of nonlinear features (shared over all recorded neurons) and fits a Poisson regression model on this shared feature space for each neuron. With this approach we effectively solve multiple nonlinear regression tasks simultaneously; hence the 'multi-task' nomenclature. The model extends simpler regression approaches by allowing nonlinear interactions between covariates. In particular, previous reduced-rank regression approaches (*Izenman, 1975*; *Kobak et al., 2016*; *Steinmetz et al., 2019*; *Posani et al., 2025*), including the analysis presented here (*Figure 7*), can be seen as a special case of the multi-task neural network, with a single hidden layer and no nonlinearity between layers.

*Figure 8b* shows model predictions on held-out trials for a single neuron in VISa/am. We plot the observed and predicted peri-event time histograms and raster plots for left trials. As a visual overview of which behavioral covariates are correlated with the MTNN prediction of this neuron's activity on each trial, the predicted raster plot and various behavioral covariates that are input into the MTNN are shown in *Figure 8c*. Overall, the MTNN approach accurately predicts the observed firing rates. When the MTNN and GLMs are trained on movement, task-related, and prior covariates, the MTNN slightly outperforms the GLMs on predicting the firing rate of held-out test trials (See *Figure 8— figure supplement 1b*). On the other hand, as shown in *Posani et al., 2025*, reduced-rank regression models can achieve similar predictive performance as the MTNN.

Next we use the predictive model performance to quantify the contribution of each covariate to the fraction of variance explained by the model. Following *Musall et al., 2019*, we run two complementary analyses to quantify these effect sizes: *single-covariate fits*, in which we fit the model using just one of the covariates, and *leave-one-out fits*, in which we train the model with one of the covariates left out and compare the predictive explained to that of the full model. As an extension of the leave-one-out analysis, we run the *leave-group-out analysis*, in which we quantify the contribution of each group of covariates (electrophysiological, task-related, and movement) to the model performance.

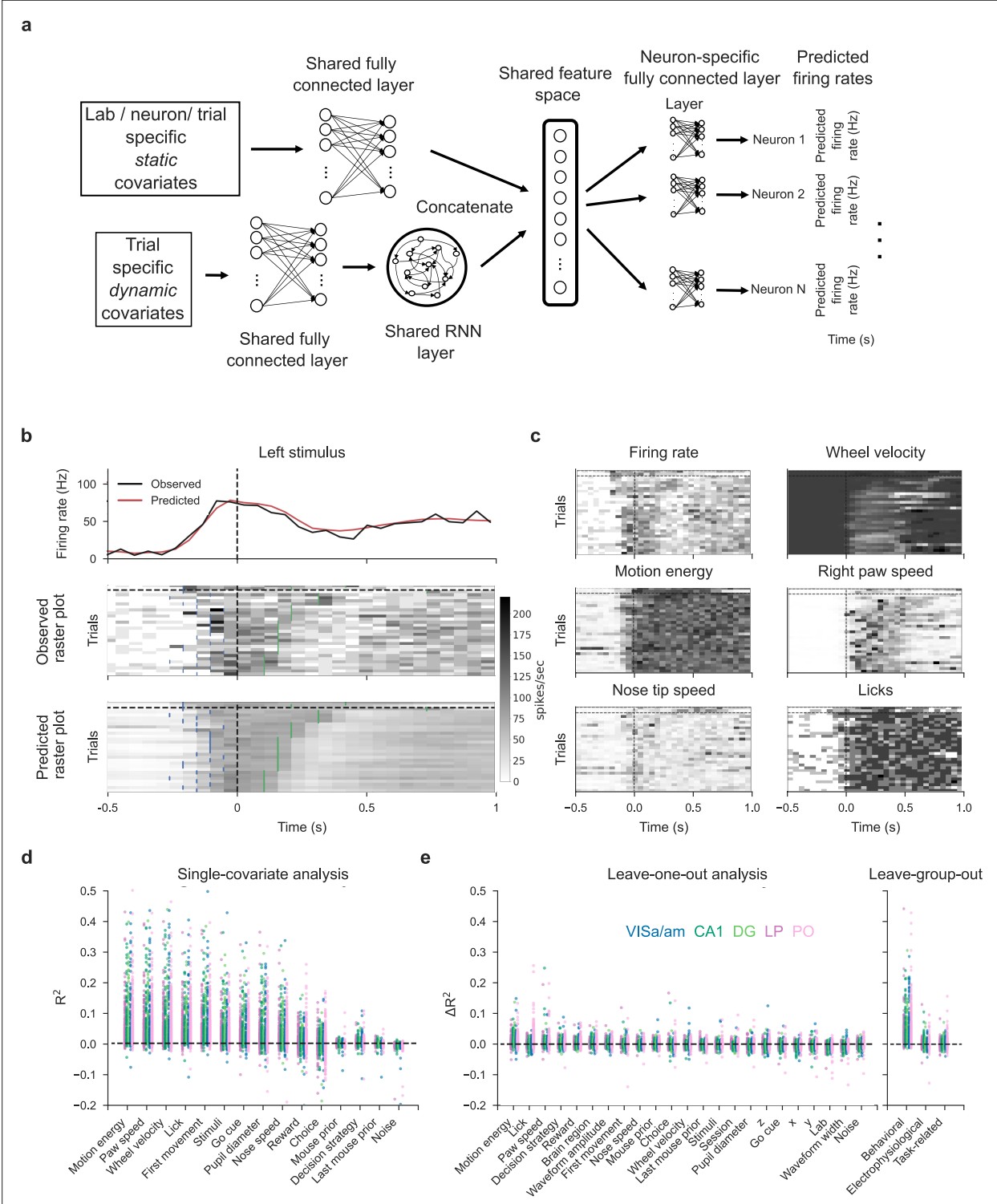

**Figure 8.** Single-covariate, leave-one-out, and leave-group-out analyses show the contribution of each (group of) covariate(s) to the MTNN model. Lab and session IDs have low contributions to the model. (**a**) We adapt a MTNN approach for neuron-specific firing rate prediction. The model takes in a set of covariates and outputs time-varying firing rates for each neuron for each trial. See **Table 2** for a full list of covariates. (**b**) MTNN model estimates of firing rates (50ms bin size) of a neuron in VISa/am from an example subject during held-out test trials. The trials with stimulus on the left are shown and are aligned to the first movement onset time (vertical dashed lines). We plot the observed and predicted PETHs and raster plots. The blue ticks in the raster plots indicate stimulus onset, and the green ticks indicate feedback times. The trials above (below) the black horizontal dashed line are incorrect (correct) trials, and the trials are ordered by reaction time. The trained model does well in predicting the (normalized) firing rates. The MTNN prediction

*Figure 8 continued on next page*

*Figure 8 continued*

quality measured in $R^2$ is 0.45 on held-out test trials and 0.94 on PETHs of held-out test trials. (**c**) We plot the MTNN firing rate predictions along with the raster plots of behavioral covariates, ordering the trials in the same manner as in (**b**). We see that the MTNN firing rate predictions are modulated synchronously with several behavioral covariates, such as wheel velocity and paw speed. (**d**) Single-covariate analysis, colored by the brain region. Each dot corresponds to a single neuron in each plot. (**e**) Leave-one-out and leave-group-out analyses, colored by the brain region. The analyses are run on 1133 responsive neurons across 32 sessions. The leave-one-out analysis shows that lab/session IDs have low effect sizes on average, indicating that within and between-lab random effects are small and comparable. The 'noise' covariate is a dynamic covariate (white noise randomly sampled from a Gaussian distribution) and is included as a negative control: the model correctly assigns zero effect size to this covariate. Covariates that are constant across trials (e.g., lab and session IDs, neuron's 3D spatial location) are left out from the single-covariate analysis.

The online version of this article includes the following figure supplement(s) for figure 8:

**Figure supplement 1.** MTNN prediction quality; MTNN slightly outperforms GLMs on predicting the firing rates of held-out test trials; PETHs and MTNN predictions for held-out test trials.

**Figure supplement 2.** MTNN prediction quality on the data simulated from GLMs is comparable to the GLMs' prediction quality.

**Figure supplement 3.** Lab IDs have a negligible effect on the MTNN prediction.

**Figure supplement 4.** MTNN single-covariate effect sizes are highly correlated across sessions and labs.

Using data simulated from GLMs, we first validate that the MTNN leave-one-out analysis is able to partition and explain different sources of neural variability (See *Figure 8—figure supplement 2*).

We then run single-covariate, leave-one-out, and leave-group-out analyses to quantify the contributions of the covariates listed in *Table 2* to the predictive performance of the model on held-out test trials. The results are summarized in *Figure 8d and e*, with a simulated positive control analysis in *Figure 8—figure supplement 3*. According to the single-covariate analysis (*Figure 8d*), face motion energy (derived from behavioral video), paw speed, wheel velocity, and first movement onset timing can individually explain about 5% of variance of the neurons on average, and these single-covariate effect sizes are highly correlated (*Figure 8—figure supplement 4*).

The leave-one-out analysis (*Figure 8e* left) shows that most covariates have low unique contribution to the predictive power. This is because many covariates are correlated and are capable of capturing variance in the neural activity even if one of the covariates is dropped (See behavioral raster plots in *Figure 8c*). According to the leave-group-out analysis (*Figure 8e* right), the 'movement' covariates as a group have the highest unique contribution to the model's performance while the task-related and electrophysiological variables have close-to-zero unique contribution. Most importantly, the leave-one-out analysis shows that lab and session IDs, conditioning on the covariates listed in *Table 2*, have close to zero effect sizes, indicating that within-lab and between-lab random effects are small and comparable.

## Decodability of task variables is consistent across labs, but varies by brain region

The previous sections addressed whether task modulation of neural activity is consistent across labs from an encoding perspective. In this section, we explored the consistency in population-level representation of task variables across labs from a decoding perspective. The analysis consists of two parts: first, we decoded selected task variables—choice, stimulus side, reward, and wheel speed—using population activity from five brain regions (PO, LP, DG, CA1, and VISa/am) across individual probe insertions. Second, we conducted statistical tests to determine whether the decodability of task variables is comparable across labs.

We used the multi-animal, across-region reduced-rank decoder introduced in *Zhang et al., 2024*. We performed fivefold cross-validation and adjusted for chance-level decoding scores for each task variable. For choice, stimulus side, and reward, we calculated chance-level accuracy by decoding based on the majority class in the training data and then subtracting this from the observed accuracy. For wheel speed, we report the single-trial $R^2$, which measures prediction quality by first calculating the trial-average behavior across choice, stimulus side, and reward conditions, and then subtracting these averages from the observed behavior. This approach adjusts for chance-level decoding by implicitly assuming that the condition-specific trial averages can serve as the chance-level prediction.

Given the decoding scores over chance level, we performed statistical tests to assess the consistency of task variable decodability across labs and regions. To assess whether there are significant

**Table 2.** List of covariates input to the multi-task neural network.

| Covariate Name | Type | Group | Note |
|---|---|---|---|
| Lab ID | Categorical / Static | | |
| Session ID | Categorical / Static | | |
| Neuron 3D spatial position | Real / Static | Electrophysiological | In standardized CCF coordinates |
| Neuron amplitude | Real / Static | Electrophysiological | Template amplitude |
| Neuron waveform width | Real / Static | Electrophysiological | Template width |
| Paw speed | Real / Dynamic | Movement | Inferred from Lightning Pose |
| Nose speed | Real / Dynamic | Movement | Inferred from DLC |
| Pupil diameter | Real / Dynamic | Movement | Inferred from Lightning Pose |
| Motion energy | Real / Dynamic | Movement | |
| Stimulus | Real / Dynamic | Task-related | Stimulus side, contrast and onset timing |
| Go cue | Binary / Dynamic | Task-related | |
| First movement | Binary / Dynamic | Task-related | |
| Choice | Binary / Dynamic | Task-related | |
| Feedback | Binary / Dynamic | Task-related | |
| Wheel velocity | Real / Dynamic | Movement | |
| Mouse Prior | Real / Static | | Mouse's prior belief |
| Last Mouse Prior | Real / Static | | Mouse's prior belief in previous trial |
| Lick | Binary / Dynamic | Movement | Inferred from DLC |
| Decision Strategy | Real / Static | | Decision-making strategy *Ashwood et al., 2021* |
| Brain region | Categorical / Static | Electrophysiological | 5 repeated site regions |

differences in mean decoding scores across different labs, we performed a one-way ANOVA test *Figure 9a*; the adjusted significance level is 0.01 after multiple-comparison correction. No statistically significant differences in decoding scores were apparent for any region or variable. This suggests that the decodability of task variables from populations is generally comparable across different labs. We then evaluated task-specific decoding performance categorized by region and lab, accounting for the number of neurons (*Figure 9b*). No clustering of decoding scores based on lab identities was apparent. We then performed a quantitative test by decoding lab and region identity using both decoding scores and the number of units as covariates. We conducted a permutation test with 5000 permutations and used a support vector classifier (again with an adjusted significance level of 0.01 after multiple-comparison correction). We could accurately determine region identity based on decoding scores and unit counts, as evidenced by macro F1 scores that exceed chance levels and significant p-values. In contrast, lab identity is less decodable than region identity, with F1 scores near chance. This implies that, after accounting for the number of neurons, the decodability of task variables from populations remains comparable across different labs.

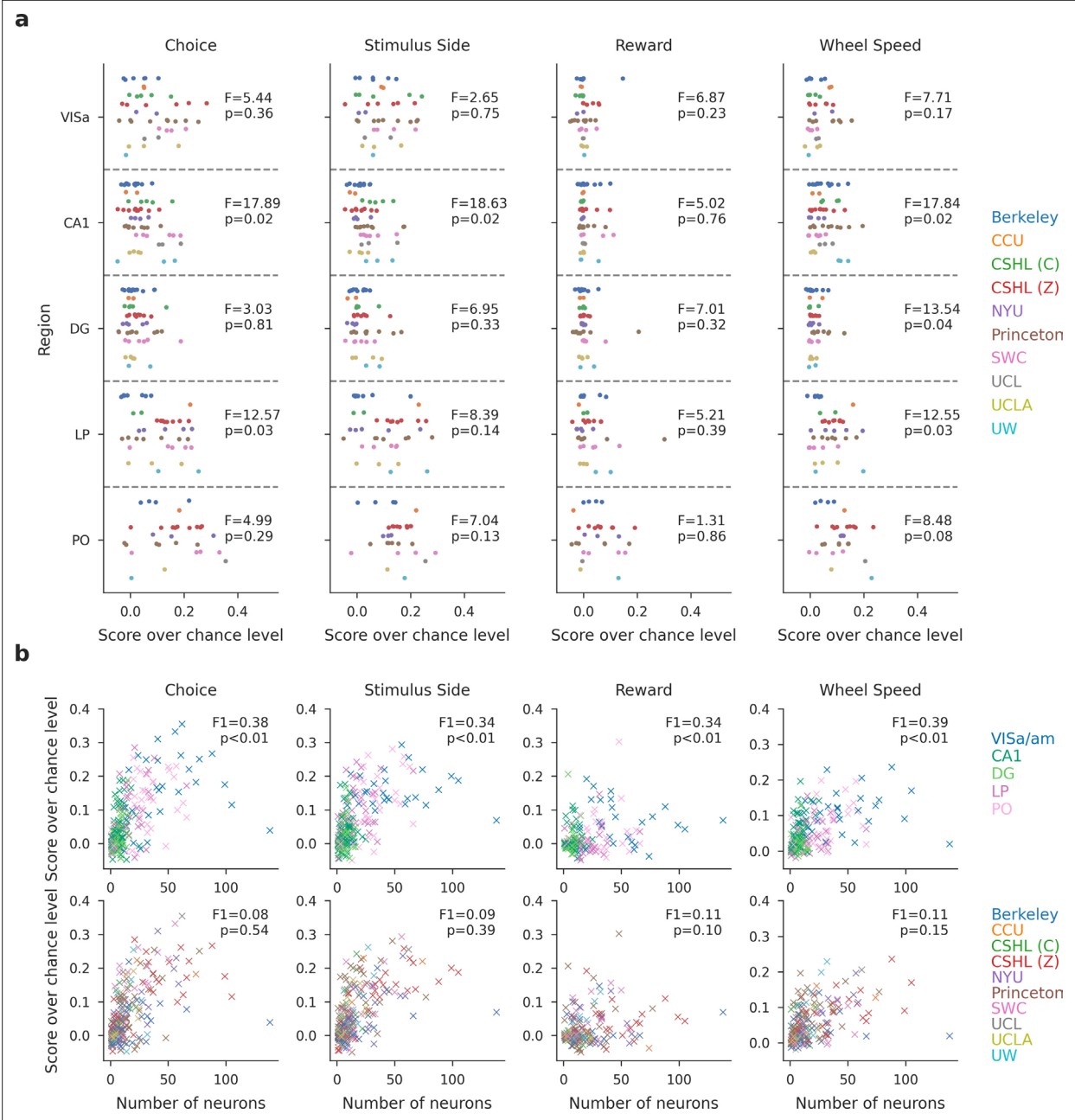

**Figure 9.** The decodability of task variables from the population is consistent across labs, but varies by brain region. (**a**) Task-specific decoding performance per region with per session results. The decoding scores over chance level for choice, stimulus side, and reward represent decoding accuracy, while the score for wheel speed is measured in $R^2$ units. Each dot indicates the decoding performance for a single insertion, color-coded by lab identity. A one-way ANOVA is conducted to determine if there are significant differences in mean decoding scores across labs. The F-statistic and p-value are reported. After multi-comparison correction, the reported p-values reveal no statistically significant differences in decoding scores across labs. (**b**) The scatter plot shows the decoding score over chance level plotted against the number of neurons used. Each dot represents a single insertion, color-coded by the region (top) and lab (bottom) identity. The scores for choice, stimulus side, and reward represent decoding accuracy, while the score for wheel speed is measured in $R^2$ units. We conduct a quantitative test of the decodability of lab and region identity using decoding scores and the number of units as covariates. A support vector classifier is used, and a permutation test with 5000 permutations is applied. We report the macro F1 scores and p-values from the permutation test. After multiple-comparison correction, lab identity is less decodable than region identity, and its observed decodability may occur by chance.

## Discussion

We set out to test whether the results from an electrophysiology experiment could be reproduced across 10 geographically separated laboratories. We observed notable variability in the position of the electrodes in the brain despite efforts to target the same location (*Figure 2*). Fortunately, after applying stringent quality-control criteria (including the RIGOR standards, *Table 1*), we found that electrophysiological features such as neuronal yield, firing rate, and LFP power were reproducible across laboratories (*Figure 3d*). The proportion of neurons with responses that were modulated, and the ways in which they were modulated by individual, behaviorally relevant task events was more variable: lab-to-lab differences were evident in some regions (*Figures 4 and 5*) and these were not explainable by, for instance, systematic variation in the subregions that were targeted (*Figure 6*). Reassuringly, analyses that summarized each neuron's activity by a compact vector of features were more robust (*Figure 7*). Further, when we trained a multi-task neural network to predict neural activity, lab identity accounted for little neural variance (*Figure 8*), arguing that the lab-to-lab differences we observed are driven by outlier neurons/sessions and nonlinear interactions between variables, rather than, for instance, systematic biases. A decoding analysis likewise confirmed reproducibility of neural activity across labs (*Figure 9*). This is reassuring, and points to the need for appropriate analytical choices to ensure reproducibility. Altogether, our results suggest that standardization and QC metrics can enhance reproducibility, and that caution should be taken with regard to electrode placement and the interpretation of some standard single-neuron metrics.

The absence of systematic differences across labs for some metrics argues that standardization of procedures is a helpful step in generating reproducible results. Our experimental design precludes an analysis of whether the reproducibility we observed was driven by person-to-person standardization or lab-to-lab standardization. Most likely, both factors contributed: all lab personnel received standardized instructions for how to implant head bars and train animals, which likely reduced personnel-driven differences. In addition, our use of standardized instrumentation and software minimized lab-to-lab differences that might normally be present. Another significant limitation of the analysis presented here is that we have not been able to assess the extent to which other choices of quality metrics and inclusion criteria might have led to greater or lesser reproducibility.

Reproducibility in our electrophysiology studies was further enhanced by rigorous QC metrics that ultimately led us to exclude a significant fraction of datasets (retaining 82 / 121 experimental sessions). QC was enforced for diverse aspects of the experiments, including histology, behavior, targeting, neuronal yield, and the total number of completed sessions. A number of recordings with high noise and/or low neuronal yield were excluded. Exclusions were driven by artifacts present in the recordings, inadequate grounding, and a decline in craniotomy health; all of these can potentially be improved with experimenter experience (a metric that could be systematically examined in future work). A few QC metrics were specific to our experiments (and thus not listed in *Table 1*). For instance, we excluded sessions with fewer than 400 trials, which could be too stringent (or not stringent enough) for other experiments.

These observations suggest that future experiments would be more consistently reproducible if researchers followed, or at least reported, a number of agreed upon criteria, such as the RIGOR standards we define in *Table 1*. This approach has been successful in other fields: for instance, the neuroimaging field has agreed upon a set of guidelines for 'best practices', and has identified factors that can impede those practices (*Nichols et al., 2017*). The genomics field likewise adopted the Minimum Information about a Microarray Experiment (MIAME) standard, designed to ensure that data from microarrays could be meaningfully interpreted and experimentally verified (*Brazma et al., 2001*). The autophagy community has a similar set of guidelines for experiments (*Klionsky, 2016*). Our work here suggests the creation of a similar set of standards for electrophysiology and behavioral experiments would be beneficial, at least on an 'opt in' basis: for instance, manuscript authors, especially those new to electrophysiology, could state which of the metrics were adhered to, and which were not needed. The metrics might also serve as the starting point for training students how to impose inclusion criteria for electrophysiological studies. Importantly, the use of RIGOR need not prevent flexibility: studies focusing on multi-unit activity would naturally omit single unit metrics, and studies on other animals or contexts would understandably need to adjust some metrics. We provide code to easily run RIGOR metrics on users' external datasets, which we hope will encourage this practice.

Establishment of such standards has the potential to enhance lab-to-lab reproducibility, but experiment-to-experiment variability may not be entirely eliminated. A large-scale effort to enhance reproducibility in *C. elegans* aging studies successfully replicated average lifespan curves across three labs by standardizing experimental methods such as handling of organisms and notation of age (e.g. when egg is hatched vs laid) (*Lithgow et al., 2017*; *Lucanic et al., 2017*). Still, variability in the lifespan curves of individual worms nevertheless persisted, warranting further studies to understand what molecular differences might explain this. Similarly, we observed no systematic difference across labs in some electrophysiological measures such as LFP power (*Figure 3d*) or functional responses in some regions but found considerable variability across experiments in other regions (*Figures 4h and 5i*).

We found probe targeting to be a large source of variability. Our ability to detect targeting error benefited from an automated histological pipeline combined with alignment and tracing that required agreement between multiple users, an approach that greatly exceeds the histological analyses done by most individual labs. Our approach, which enables scalability and standardization across labs while minimizing subjective variability, revealed that much of the variance in targeting was due to the probe entry positions at the brain surface, which were randomly displaced across the dataset. The source of this variance could be due to a discrepancy in skull landmarks compared to the underlying brain anatomy. Accuracy in placing probes along a planned trajectory is therefore limited by this variability (about 400μm). Probe angle also showed a small degree of variance and a bias in both anterior-posterior and medio-lateral direction, indicating that the Allen CCF (*Wang et al., 2020*) and stereo-taxic coordinate systems are slightly offset. Detecting this offset relied on a large cohort size and an automated histological pipeline, but now that we have identified the offset, it can be easily accounted for by any lab. Specifically, probe angles must be carefully computed from the CCF, as the CCF and stereotaxic coordinate systems do not define the same coronal plane angle. Minimizing variance in probe targeting is another important element in increasing reproducibility, as slight deviations in probe entry position and angle can lead to samples from different populations of neurons. Collecting structural MRI data in advance of implantation (*Browning, 2023*) could reduce targeting error, although this is infeasible for most labs. A more feasible solution is to rely on stereotaxic coordinates but account for the inevitable off-target measurements by increasing cohort sizes and adjusting probe angles when blood vessels obscure the desired location. This is essential because small differences in probe location may be responsible for other studies arriving at different conclusions, highlighting the need for agreed upon methods for targeting specific areas (*Rajasethupathy et al., 2015*; *Andrianova et al., 2022*).

Our results also highlight the critical importance of reproducible histological processing and subsequent probe alignment. Specifically, we used a centralized histology and registration pipeline to assign each recording site on each probe to a particular anatomical location, based on registration of the histological probe trajectories to the CCF and the electrophysiological features recorded at each site. This differs from previous approaches, in which stereotaxic coordinates alone were used to target an area of interest and exclusion criteria were not specified; see for example *Harvey et al., 2012*; *Raposo et al., 2014*; *Erlich et al., 2015*; *Goard et al., 2016*; *Najafi et al., 2020*. The reliance on stereotaxic coordinates for localization, instead of standardized histological registration, is a possible explanation for conflicting results across laboratories in previous literature. Our results speak to the importance of adopting standardized procedures more broadly across laboratories. Notably, central-izing the histology pipeline likely reduces variance, but the extent of additional variability introduced by individual lab implementations remains unclear.

A major contribution of our work is open-source data and code: we share our full dataset (see Data Availability) and suite of analysis tools for quantifying reproducibility (see Code Availability) and computing the RIGOR standards. The analyses here required significant improvements in data archi-tecture, visualization, spike sorting, histology image analysis, and video analysis. Our analyses uncov-ered major gaps and issues in the existing toolsets that required improvements (see Materials and methods and *Banga et al., 2022*; *Birman et al., 2022* for full details). For example, we improved existing spike sorting pipelines with regard to scalability, reproducibility, and stability. These improve-ments contribute toward advancing automated spike sorting, and move beyond subjective manual curation, which scales poorly and limits reproducibility. We anticipate that our open-source dataset will play an important role in further improvements to these pipelines and also the development

of further methods for modeling the spike trains of many simultaneously recorded neurons across multiple brain areas and experimental sessions.

Scientific advances rely on the reproducibility of experimental findings. The current study demonstrates that reproducibility is attainable for many features measured during a standardized perceptual decision task. We offer several recommendations to enhance reproducibility, including (1) standardized protocols for data collection and processing, (2) methods to account for variability in electrode targeting, and (3) rigorous data quality metrics, such as RIGOR. These recommendations are urgently needed as neuroscience continues to move toward increasingly large and complex datasets.

## Resources
### Code availability
All code is freely available. Please visit https://github.com/int-brain-lab/paper-reproducible-ephys (copy archived at *Meijer et al., 2024*) to access the code used to produce the results and figures presented in this article. The RIGOR metrics can be computed on external datasets by following the tutorial at https://github.com/int-brain-lab/paper-reproducible-ephys/blob/master/RIGOR_script.ipynb.

### Protocols and pipelines
Please visit https://figshare.com/projects/Reproducible_Electrophysiology/138367 to access the protocols and pipelines used in this article.

### Quality control and data inclusion
Please see this spreadsheet for a comprehensive overview of which recordings and the number of cells, mice and sessions that are used in each figure panel, as well as the reasons for inclusion or exclusion.

# Materials and methods
All procedures and experiments were carried out in accordance with local laws and following approval by the relevant institutions: the Animal Welfare Ethical Review Body of University College London; the Institutional Animal Care and Use Committees of Cold Spring Harbor Laboratory, Princeton University, University of California at Los Angeles, and University of California at Berkeley; the University Animal Welfare Committee of New York University; and the Portuguese Veterinary General Board.

## Animals
Mice were housed under a 12/12 hr light/dark cycle (normal or inverted depending on the laboratory) with food and water available ad libitum, except during behavioral training days. Electrophysiological recordings and behavioral training were performed during either the dark or light phase of the cycle depending on the laboratory. N=78 adult mice (C57BL/6, male and female, obtained from either Jackson Laboratory or Charles River) were used in this study. Mice were aged 111 - 442 days and weighed 16.4-34.5 g on the day of their first electrophysiology recording session in the repeated site.

## Materials and apparatus
Briefly, each lab installed a standardized electrophysiological rig (named 'ephys rig' throughout this text), which differed slightly from the apparatus used during behavioral training (*Aguillon-Rodriguez et al., 2021*). The general structure of the rig was constructed from Thorlabs parts and was placed inside a custom acoustical cabinet clamped on an air table (Newport, M-VIS3036-SG2-325A). A static head bar fixation clamp and a 3D-printed mouse holder were used to hold a mouse such that its forepaws rest on the steering wheel (86652 and 32019, LEGO; *Aguillon-Rodriguez et al., 2021*). Silicone tubing controlled by a pinch valve (225P011-21, NResearch) was used to deliver water rewards to the mouse. The display of the visual stimuli occurred on a LCD screen (LP097Q × 1, LG). To measure the precise times of changes in the visual stimulus, a patch of pixels on the LCD screen flipped between white and black at every stimulus change, and this flip was captured with a photodiode (Bpod Frame2TTL, Sanworks). Ambient temperature, humidity, and barometric air pressure were measured

with the Bpod Ambient module (Sanworks), wheel position was monitored with a rotary encoder (05.2400.1122.1024, Kubler).

Videos of the mouse were recorded from three angles (left, right, and body) with USB cameras (CM3-U3-13Y3M-CS, Point Grey). The left camera acquires at 60Hz; full resolution (1280 x1024), right camera at 150Hz; half resolution (640x512), and body camera at 30Hz; half resolution (640Hzx512). A custom speaker (Hardware Team of the Champalimaud Foundation for the Unknown, V1.1) was used to play task-related sounds, and an ultrasonic microphone (Ultramic UM200K, Dodotronic) was used to record ambient noise from the rig. All task-related data was coordinated by a Bpod State Machine (Sanworks). The task logic was programmed in Python and the visual stimulus presentation and video capture were handled by Bonsai (*Lopes et al., 2015*) utilizing the Bonsai package BonVision (*Lopes et al., 2021*).

All recordings were made using Neuropixels probes (Imec, 3A and 3B models), advanced in the brain using a micromanipulator (Sensapex, uMp-4) tilted by a 15 degree angle from the vertical line. The aimed electrode penetration depth was 4.0 mm. Data were acquired via an FPGA (for 3A probes) or PXI (for 3B probes, National Instrument) system and stored on a PC.

## Headbar implant surgery

Mice were placed in an induction box with 3-4% isoflurane and maintained at 1.5-2% isoflurane. Saline 10mg/kg subcutaneously is given each hour. The mouse is placed in the stereotaxic frame using ear bars placed in the ridge posterior to the ear canal. The mouse is then prepped for surgery, removing hair from the scalp using epilation creme. Much of the underlying periosteum was removed and bregma and lambda were marked. Then the head was positioned such that there was a 0 degree angle between bregma and lambda in all directions. Lateral and middle tendons are removed using fine forceps. The head bar was then placed in one of three stereotactically defined locations and cemented in place These locations are: AP -6.90, ML +/- 1.25(curved headbar placed caudally onto cerebellum), AP +1.36, ML +/- 1.25 (curved headbar placed rostrally onto frontal zones), and AP -2.95, ML +/-1.25 (straight headbar placed centrally). The location of planned future craniotomies were measured using a pipette referenced to bregma and marked on the skull using either a surgical blade or a pen. A small amount of vetbond was applied to the edges of the skin wound to seal it off and create more surface area.The exposed skull was then covered with cement and clear UV curing glue, ensuring that the remaining scalp was unable to retract from the implant.

## Behavioral training and habituation to the ephys rig

All recordings performed in this study were done in expert mice. To reach this status, animals were habituated for 3 days and trained for several days in the equal probability task version where the Gabor patch appears on the right or left side of the screen with equal probability. Animals are trained to move the visual stimulus controlled by a wheel toward the center of the screen. Animals must reach a 'trained 1b' status wherein each of the three consecutive sessions, the mouse completed over 400 trials and performed over 90% on the easy (contrast >= 50%) trials. Additionally, the median reaction time across these sessions must be below 2 seconds for the 0% contrast. Lastly, a psychometric curve is fitted with four parameters bias, lapse right, lapse left and threshold, must meet the following criteria: the absolute bias must be below 10, the threshold below 20, and each lapse below 0.1. Once these conditions are met, animals progress to 'biasedChoiceWorld' in which they are first presented with an unbiased block of trials and subsequently blocks are from either of two biased blocks: Gabor patch is presented on the left and right with probabilities of 0.2 and 0.8 (20:80) respectively, and in the other block type the Gabor patch is presented on the left and right with probabilities of 0.8 and 0.2 (80:20), respectively. In summary, once mice learned the biasedChoiceWorld task (criteria 'ready4ephysRig' reached), they were habituated to the electrophysiology rig. Briefly, this criterion is met by performing three consecutive sessions that meet 'trained 1b' status. Additionally, psychometric curves (separately fit for each block type) must have bias shifts < 5%, and lapse rates measured on asymmetric blocks must be below 0.1. Their first requirement was to perform one session of biasedChoiceWorld on the electrophysiology rig, with at least 400 trials and 90% correct on easy contrasts (collapsing across block types). Once this criterion was reached, time delays were introduced at the beginning of the session; these delays served to mimic the time it would take to insert electrodes in the brain. To be included in subsequent sessions, mice were required to maintain performance for

three subsequent sessions (same criterion as 'ready4ephysRig'), with a minimum of one session with a 15-min pre-session delay. For the analyses in this study, only electrophysiology sessions where the mouse completed at least 400 trials were used.

## Electrophysiological recording using Neuropixels probes
### Data acquisition
Briefly, upon the day of electrophysiological recording, the animal was anesthetized using isoflurane and surgically prepared. The UV glue was removed using ethanol and a biopsy punch or a scalpel blade. The exposed skull was then checked for infection. A test was made to check whether the implant could hold liquid without leaking to ensure that the brain did not dry during the recording. Subsequently, a grounding pin was cemented to the skull using Metabond. 1-2 craniotomies (1 × 1 mm) were made over the marked locations using a biopsy punch or drill. The dura was left intact, and the brain was lubricated with ACSF. DuraGel was applied over the dura as a moisturising sealant, and covered with a layer of Kwikcast. The mouse was administered analgesics subcutaneously, and left to recover in a heating chamber until locomotor and grooming activity were fully recovered.

Once the animal was recovered from the craniotomy, it was relocated to the apparatus. Once a craniotomy was made, up to four subsequent recording sessions were made in that same craniotomy. Up to two probes were implanted in the brain on a given session.

### Probe track labeling
CM-DiI (V22888, Thermofisher) was used to label probes for subsequent histology. CM-DiI was strored in the freezer -at 20C until ready for use. On the day of recording, we thawed CM-DiI at room temperature, protecting it from light. Labeling took place under a microscope while the Neuropixels probe was secured onto a micromanipulator, electrode sites facing up. 1uL of CM-DiI was placed onto either a coverslip or parafilm. Using the micromanipulator, the probe tip was inserted into the drop of dye with care taken to not get dye onto the electrode sites. For Neuropixels probes, the tip extends about 150um from the first electrode site. The tip is kept in the dye until the drop dries out completely (approximately 30 s) and then the micromanipulator is slowly retracted to remove the probe.

### Spike sorting
Raw electrophysiological recordings were initially saved in a flat uncompressed binary format, representing a storage of 1.3 GB/min. To save disk space and achieve better transfer speeds,we utilized simple lossless compression to achieve a compression ratio between 2x and 3x. In many cases, we encounter line noise due to voltage leakage on the probe. This translates into large 'stripes' of noise spanning the whole probe. To reduce the impact of these noise 'stripes', we perform three main preprocessing steps including: (1) correction for 'sample shift' along the length of the probe by aligning the samples with a frequency domain approach; (2) automatic detection, rejection and interpolation of failing channels; (3) application of a spatial 'de-striping' filter. After these preprocessing steps, spike sorting was performed using ibl-sorter, a Python port of the Kilosort 2.5 algorithm that includes modifications to preprocessing (*Steinmetz et al., 2021*). At this step, we apply registration, clustering, and spike deconvolution. We found it necessary to improve the original code in several aspects (e.g. improved modularity and documentation, and better memory handling for datasets with many spikes) and developed an open-source Python port; the code repository is here: *Banga, 2024*. See *Banga et al., 2022* for full details.

### Single cluster quality metrics
To determine whether a single cluster will be used in downstream analysis, we used three metrics (listed as part of RIGOR in *Table 1*): the refractory period, an amplitude cut-off estimate, and the median of the amplitudes. First, we developed a metric which estimates whether a neuron is contaminated by refractory period violations (indicating potential overmerge problems in the clustering step) without assuming the length of the refractory period. For each of the many refractory period lengths, we compute the number of spikes (refractory period violations) that would correspond to some maximum acceptable amount of contamination (chosen as 10%). We then compute the likelihood of observing fewer than this number of spikes in that refractory period under the assumption of Poisson

spiking. For a neuron to pass this metric, this likelihood that our neuron is less than 10% contaminated, must be larger than 90% for any one of the possible refractory period lengths.

Next, we compute an amplitude cut-off estimate. This metric estimates whether an amplitude distribution is cut-off by thresholding in the deconvolution step (thus leading to a large fraction of missed spikes). To do so, we compare the lowest bin of the histogram (the number of neurons with the lowest amplitudes), to the bins in the highest quantile of the distribution (defined as the top 1/4 of bins higher than the peak of the distribution.) Specifically, we compute how many SDs the height of the low bin falls outside of the mean of the height of the bins in the high quantile. For a neuron to pass this metric, this value must be less than 5 SDs, and the height of the lowest bin must be less than 10% of the height of the peak histogram bin. See [ref] for further details.

Finally, we compute the median of the amplitudes. For a neuron to pass this metric, the median of the amplitudes must be larger than 50 uV.

## Local field potential (LFP)

Concurrently with the action potential band, each channel of the Neuropixel probe recorded a low-pass filtered trace at a sampling rate of 2500 Hz. A denoising was applied to the raw data, comprising four steps. First a Butterworth low-cut filter is applied on the time traces, with 2Hz corner frequency and order 3. Then a subsample shift was applied to rephase each channel according to the time-sampling difference due to sequential sampling of the hardware. Then faulty bad channels were automatically identified, removed, and interpolated. Finally, the median reference is subtracted at each time sample. See *Banga et al., 2022* for full details. After this processing, the power spectral density at different frequencies was estimated per channel using the Welch's method with partly overlapping Hanning windows of 1024 samples. Power spectral density (PSD) was converted into dB as follows:

$$dB = 10 * log(PSD) \tag{1}$$

## Serial section two-photon imaging

Mice were given a terminal dose of pentobarbital intraperitoneally. The toe-pinch test was performed as confirmation that the mouse was deeply anesthetized before proceeding with the surgical procedure. Thoracic cavity was opened, the atrium was cut, and PBS followed by 4% formaldehyde solution (Thermo Fisher 28908) in 0.1M PB pH 7.4 were perfused through the left ventricle. The whole mouse brain was dissected and post-fixed in the same fixative for a minimum of 24 hr at room temperature. Tissues were washed and stored for up to 2–3 weeks in PBS at 4°C, prior to shipment to the Sainsbury Wellcome Centre for image acquisition.

The brains were embedded in agarose and imaged in a water bath filled with 50 mM PB using a 4 kHz resonant scanning serial section two-photon microscopy (*Economo et al., 2016*; *Ragan et al., 2012*). The microscope was controlled with ScanImage Basic (Vidrio Technologies, USA), and Baking-Tray, a custom software wrapper for setting up the imaging parameters (*Campbell, 2020*). Image tiles were assembled into 2D planes using StitchIt (*Campbell, 2021*). Whole brain coronal image stacks were acquired at a resolution of 4.4 x 4.4 x 25.0 µm in XYZ (Nikon 16x NA 0.8), with a two-photon laser wavelength of 920 nm, and approximately 150 mW at the sample. The microscope cut 50 µm sections using a vibratome (Leica VT1000) but imaged two optical planes within each slice at depths of about 30 µm and 55 µm from the tissue surface using a PIFOC. Two channels of image data were acquired simultaneously using Hamamatsu R10699 multialkali PMTs: 'Green' at 525 nm ±25 nm (Crhoma ET525/50m); 'Red' at 570 nm low pass (Chroma ET570lp).

Whole brain images were downsampled to 25 µm isotropic voxels and registered to the adult mouse Allen CCF (*Wang et al., 2020*) using BrainRegister (*West, 2022*), an elastix-based (*Klein et al., 2010*) registration pipeline with optimized parameters for mouse brain registration. Two independent registrations were performed, samples were registered to the CCF template image and the CCF template was registered to the sample.

## Probe track tracing and alignment

Tracing of Neuropixels electrode tracks was performed on registered image stacks. Neuropixels probe tracks were manually traced to yield a probe trajectory using Lasagna (*Campbell et al., 2020*), a Python-based image viewer equipped with a plugin tailored for this task. Tracing was performed on

the merged images on the green (auto-fluorescence) and red (CM-Dil labeling) channels, using both coronal and sagittal views. Traced probe track data was uploaded to an Alyx server *Rossant et al., 2022*; a database designed for experimental neuroscience laboratories. Neuropixels channels were then manually aligned to anatomical features along the trajectory using electrophysiological landmarks with a custom electrophysiology alignment tool (*Faulkner, 2020*; *Liu et al., 2021*). This alignment process was performed by the experimenter and an additional member to ensure agreement on the assigned channel locations.

## Permutation tests and power analysis

We use permutation tests to study the reproducibility of neural features across laboratories. To this end, we first defined a test statistic that is sensitive to systematic deviations in the distributions of features between laboratories: the maximum absolute difference between the cumulative distribution function (CDF) of a neural feature within one lab and the CDF across all other labs (similar to the test statistic used for a Kolmogorov–Smirnov test). For the CDF, each mouse might contribute just a single value (e.g. in the case of the deviations from the target region), or a number for every neuron in that mouse (e.g. in the case of comparing firing rate differences during specific time-periods). The deviations between CDFs from all the individual labs are then reduced into one number by considering only the deviation of the lab with the strongest such deviation, giving us a metric that quantifies the difference between lab distributions. The null hypothesis is that there is no difference between the different laboratory distributions, that is the assignment of mice to laboratories is completely random. We sampled from the corresponding null distribution by permuting the assignments between laboratories and mice randomly 50,000 times (leaving the relative numbers of mice in laboratories intact) and computing the test statistic on these randomized samples. Given this sampled null distribution, the p-value of the permutation test is the proportion of the null distribution that has more extreme values than the test statistic that was computed on the real data.

For the power analysis (*Figure 4*, *Figure 4—figure supplement 2*), the goal was to find how much each value (firing rate modulations or Fano factors) would need to shift within the individual labs to create a significant p-value for any given test. This grants us a better understanding of the workings and limits of our test. As we chose an $\alpha$ level of 0.01 (implementing a Bonferroni correction for the number of regions, but not the number of tests, so as to not have too lenient a criterion), we needed to find the perturbations that gave a p-value <0.01. To achieve this for a given test and a given lab, we took the values of every neuron within that lab, and shifted them all up or down by a certain amount. We used binary search to find the exact points at which such an up- or down-shift caused the test to become significant. This analysis tells us exactly at which points our test becomes significant, and importantly, ensures that our permutation test is sensitive enough to detect deviations of a given magnitudes. It may seem counter intuitive that some tests allow for larger deviations than others, or that even within the same test some labs have a different range of possible perturbations than others. This is because the test considers the entire distribution of values, resulting in possibly complex interactions between the labs. Precisely because of these interactions of the data with the test, we performed a thorough power analysis to ensure that our procedure is sufficiently sensitive to across-lab variations. The bottom row of *Figure 4*, *Figure 4—figure supplement 2* shows the overall distribution of permissible shifts, the large majority of which is below one SD of the corresponding lab distribution.

## Dimensionality reduction of PETHs via principal component analysis

The analyses in *Figure 5* rely on principal component analysis (PCA) to embed PETHs into a two-dimensional feature space. Our overall approach is to compute PETHs, split into fast-reaction-time and slow-reaction-time trials, then concatenate these PETH vectors for each cell to obtain a summary of each cell's activity. Next, we stack these double PETHs from all labs into a single matrix. Finally, we used PCA to obtain a low-rank approximation of this PETH matrix.

In detail, the two PETHs consist of one averaging fast reaction time ($< 0.15 sec$, $0.15 sec$ being the mean reaction time when considering the distribution of reaction times across all sessions) trials and the other slow reaction time ($> 0.15 sec$) trials, each of length $T$ time steps. We used $20\,ms$ bins, from $-0.5\,sec$ to $1.5\,sec$ relative to motion onset, so $T = 100$. We also performed a simple normalization on each PETH, subtracting baseline activity and then dividing the firing rates by the baseline firing rate

(prior to motion onset) of each cell plus a small positive offset term (to avoid amplifying noise in very low-firing cells), following *Steinmetz et al., 2021*.

Let the stack of these double PETH vectors be $Y$, being a $N \times 2T$ matrix, where $N$ is the total number of neurons recorded across five brain regions and labs. Running principal components analysis (PCA) on $Y$ (singular value decomposition) is used to obtain the low-rank approximation $UV \approx Y$. This provides a simple low-d embedding of each cell: $U$ is $N \times k$, with each row of $U$ representing a $k$-dimensional embedding of a cell that can be visualized easily across labs and brain regions. $V$ is $k \times 2T$ and corresponds to the $k$ temporal basis functions that PCA learns to best approximate $Y$. *Figure 5a* shows the responses of two cells of $Y$ (black traces) and the corresponding PCA approximation from $UV$ (red traces).

The scatter plots in *Figure 5c, f* show the embedding $U$ across labs and brain regions, with the embedding dimension $k = 2$. Each $k \times 1$ vector in $U$, corresponding to a single cell, is assigned to a single dot in *Figure 5c, f*.

## Linear regression model to quantify the contribution of spatial and spike features to variability

To fit a linear regression model to the session-averaged firing rate of neurons, for each brain region, we used a Nx5 predictor matrix where N is the number of recorded neurons within the region. The five columns contain the following five covariates for each neuron: x, y, z position, spike amplitude, and spike peak-to-trough duration. The Nx1 observation matrix consisted of the average firing rate for each neuron throughout the entire recording period. The linear model was fit using ordinary least-squares without regularization. The unadjusted coefficient of determination ($R^2$) was used to report the level of variability in neuronal firing rates explained by the model.

## Video analysis

In the recording rigs, we used three cameras, one called 'left' at full resolution (1280x1024) and 60 Hz filming the mouse from one side, one called 'right' at half resolution (640x512) and 150 Hz, filming the mouse symmetrically from the other side, and one called 'body' filming the trunk of the mouse from above. Several QC metrics were developed to detect video issues such as poor illumination or accidental misplacement of the cameras.

We used DeepLabCut (*Mathis et al., 2018*) to track various body parts such as the paws, nose, tongue, and pupil. The pipeline first detects four regions of interest (ROI) in each frame, crops these ROIs using ffmpeg (*Tomar, 2006*) and applies a separate network for each ROI to track features. For each side video, we track the following points:

- ROI eye:

  'pupil_top_r', 'pupil_right_r', 'pupil_bottom_r', 'pupil_left_r'

- ROI mouth:

  'tongue_end_r', 'tongue_end_l'

- ROI nose:

  'nose_tip'

- ROI paws:

  'paw_r', 'paw_l'

The right side video was flipped and spatially up-sampled to look like the left side video, such that we could apply the same DeepLabCut networks. The code is available here: (*International Brain Laboratory, 2022*).

Extensive curating of the training set of images for each network was required to obtain reliable tracking across animals and laboratories. We annotated in total more than 10K frames, across

several iterations, using a semi-automated tracking failure detection approach, which found frames with temporal jumps, three-dimensional re-projection errors when combining both side views, and heuristic measures of spatial violations. These selected 'bad' frames were then annotated and the network re-trained. To find further raw video and DeepLabCut issues, we inspected trial-averaged behaviors obtained from the tracked features, such as licking aligned to feedback time, paw speed aligned to stimulus onset and scatter plots of animal body parts across a session superimposed onto example video frames. See *International Brain Laboratory, 2022*; *Aguillon-Rodriguez et al., 2021* for full details.

Despite the large labeled dataset and multiple network retraining iterations described above, DeepLabCut was not able to achieve sufficiently reliable tracking of the paws or pupils. Therefore, we used an improved tracking method, Lightning Pose, for these body parts (*Biderman et al., 2024*). Lightning Pose models were trained on the same final labeled dataset used to train DeepLabCut. We then applied the Ensemble Kalman Smoother (EKS) post-processor introduced in *Biderman et al., 2024*, which incorporates information across multiple networks trained on different train/validation splits of the data. For the paw data, we used a version of EKS that further incorporates information across the left and right camera views in order to resolve occlusions. See *Biderman et al., 2024* for full details.

## Multi-task neural network model to quantify sources of variability

### Data preprocessing

For the multi-task neural network (MTNN) analysis, we used data from 32 sessions recorded in SWC, CCU, CSHL (C), UCL, Berkeley, NYU, UCLA, and UW. We filtered out the sessions with unreliable behavioral traces from video analysis and selected labs with at least four sessions for the MTNN analysis. For the labs with more than four sessions, we selected sessions with more recorded neurons across all repeated sites. We included various covariates in our feature set (e.g. go-cue signals, stimulus/reward type, DeepLabCut/Lightning Pose behavioral outputs). For the 'decision strategy' covariate, we used the posterior estimated state probabilities of the four-state GLM-HMMs trained on the sessions used for the MTNN analysis (*Ashwood et al., 2021*). Both biased and unbiased data were used when training the four-state model. For each session, we first filtered out the trials where no choice is made. We then selected the trials whose stimulus onset time is no more than 0.4 s before the first movement onset time and feedback time is no more than 0.9 s after the first movement onset time. Finally, we selected responsive neurons whose mean firing rate is greater than 5 spikes/s for further analyses, to make model training more efficient; using a lower threshold of 1 Hz did not substantially change the results.

The lab IDs and session IDs were each encoded in a 'one-hot' format (i.e. each lab is encoded as a length 8 one-hot vector). For the leave-one-out effect size of the session IDs, we compared the model trained with all of the covariates in *Table 2* against the model trained without the session IDs. For the leave-one-out effect size of the lab IDs, we compared the model trained without the lab IDs against the model trained without both the lab and session IDs. We prevented the lab and session IDs from containing overlapping information with this encoding scheme, where the lab IDs cannot be predicted from the session IDs, and vice versa, during the leave-one-out analysis.

### Model architecture

Given a set of covariates in *Table 2*, the MTNN predicts the target sequence of firing rates from 0.5 s before first movement onset to 1 second after, with bin width set to 50 ms (30 time bins). More specifically, a sequence of feature vectors $x_{\text{dynamic}} \in \mathbb{R}^{D_{\text{dynamic}} \times T}$ that include dynamic covariates, such as Deep Lab Cut (DLC) outputs, and wheel velocity, and a feature vector $x_{\text{static}} \in \mathbb{R}^{D_{\text{static}}}$ that includes static covariates, such as the lab ID, neuron's 3-D location, are input to the MTNN to compute the prediction $y^{pred} \in \mathbb{R}^T$, where $D_{\text{static}}$ is the number of static features, $D_{\text{dynamic}}$ is the number of dynamic features, and $T$ is the number of time bins. The MTNN has initial layers that are shared by all neurons, and each neuron has its designated final fully-connected layer.

Given the feature vectors $x_{\text{dynamic}}$ and $x_{\text{static}}$ for session $s$ and neuron $u$, the model predicts the firing rates $y^{pred}$ by:

$$e_{\text{static}} = f(w_{\text{static}}^T x_{\text{static}} + b_{\text{static}}) \tag{2}$$

$$e_{\text{dynamic}} = f(w_{\text{dynamic}}^T x_{\text{dynamic}} + b_{\text{dynamic}}) \tag{3}$$

$$h_t^{(forward)} = max(0, U_1 e_{\text{dynamic},t} + V_1 h_{t-1}^{(forward)} + b_{forward}) \tag{4}$$

$$h_t^{(backward)} = max(0, U_2 e_{\text{dynamic},t} + V_2 h_{t+1}^{(backward)} + b_{backward}) \tag{5}$$

$$y_t^{pred} = f(w_{(s,u)}^T \text{concat}(e_{\text{static}}, h_t^{(forward)}, h_t^{(backward)}) + b_{(s,u)}) \tag{6}$$

where $f$ is the activation function. *Equation 2* and *Equation 3* are the shared fully-connected layers for static and dynamic covariates, respectively. *Equation 4* and *Equation 5* are the shared one-layer bidirectional recurrent neural networks (RNNs) for dynamic covariates, and *Equation 6* is the neuron-specific fully-connected layer, indexed by $(s, u)$. Each part of the MTNN architecture can have an arbitrary number of layers. For our analysis, we used two fully-connected shared layers for static covariates (*Equation 2*) and four-layer bidirectional RNNs for dynamic covariates, with the embedding size set to 128.

## Model training

The model was implemented in PyTorch and trained on a single GPU. The training was performed using Stochastic Gradient Descent on the Poisson negative loglikelihood (Poisson NLL) loss with learning rate set to 0.1, momentum set to 0.9, and weight decay set to $10^{-15}$. We used a learning rate scheduler such that the learning rate for the $i$-th epoch is $0.1 \times 0.95^i$, and the dropout rate was set to 0.15. We also experimented with mean squared error (MSE) loss instead of Poisson NLL loss, and the results were similar. The batch size was set to 1024.

The dataset consists of 32 sessions, 1133 neurons and 11490 active trials in total. For each session, 20% of the trials are used as the test data and the remaining trials are split 20:80 for the validation and training sets. During training, the performance on the held-out validation set is checked after every three passes through the training data. The model is trained for 100 epochs, and the model parameters with the best performance on the held-out validation set are saved and used for predictions on the test data.

## Simulated experiments

For the simulated experiment in *Figure 8—figure supplement 2*, we first trained GLMs on the same set of 1133 responsive neurons from 32 sessions used for the analysis in *Figure 8d and e*, with a reduced set of covariates consisting of stimulus timing, stimulus side and contrast, first movement onset timing, feedback type and timing, wheel velocity, and mouse's priors for the current and previous trials. The kernels of the trained GLMs show the contribution of each of the covariates to the firing rates of each neuron. For each simulated neuron, we used these kernels of the trained GLM to simulate its firing rates for 500 randomly initialized trials. The random trials were 1.5 seconds long with 50 ms bin width. For all trials, the first movement onset timing was set to 0.5 second after the start of the trial, and the stimulus contrast, side, onset timing and feedback type, timing were randomly sampled. We used wheel velocity traces and mouse's priors from real data for simulation. We finally ran the leave-one-out analyses with GLMs/MTNN on the simulated data and compared the effect sizes estimated by GLMs and MTNN.

## Diversity statement

We support inclusive, diverse and equitable conduct of research. One or more of the authors of this paper self-identifies as a member of an underrepresented ethnic minority in science. One or more of the authors self-identifies as a member of the LGBTQIA+ community.

## Acknowledgements

This work was supported by grants from the Wellcome Trust (209558 and 216324), National Institutes of Health (1F32MH123010, 1U19NS123716, including a Diversity Supplement) and the Simons Collaboration on the Global Brain. We thank R Poldrack, T Zador, P Dayan, S Hofer, N Ghani, and C Hurwitz for helpful comments on the manuscript. We thank Anna li for graphical contributions to *Figures 1 and 3*. The production of all IBL Platform Papers is led by a Task Force, which defines the scope and composition of the paper, assigns and/or performs the required work for the paper, and

ensures that the paper is completed in a timely fashion. The Task Force members for this platform paper include authors SAB, GC, AKC, MFD, CL, HDL, MF, GM, LP, NR, MS, NAS, MT, and SJW.

## Additional information

### Competing interests

International Brain Laboratory: Anne K Churchland: receives an honorarium from the Simons Foundation as part of her role as a member of the scientific advisory committee. Jean-Paul Noel: Reviewing editor, *eLife*. The other authors declare that no competing interests exist.

### Funding

| Funder | Grant reference number | Author |
|---|---|---|
| Simons Foundation | Simons Collaboration on the Global Brain | International Brain Laboratory |
| Wellcome Trust | 10.35802/216324 | International Brain Laboratory |
| National Institutes of Health | U19NS123716 | International Brain Laboratory |
| Wellcome Trust | 10.35802/209558 | International Brain Laboratory |
| National Institutes of Health | 1F32MH123010 | International Brain Laboratory |

The funders had no role in study design, data collection and interpretation, or the decision to submit the work for publication. For the purpose of Open Access, the authors have applied a CC BY public copyright license to any Author Accepted Manuscript version arising from this submission.

### Author contributions

International Brain Laboratory, Is the consortium that led this effort; Kush Banga, Software, Methodology; Julius Benson, Investigation; Jai Bhagat, Data curation, Software, Formal analysis, Validation, Visualization, Methodology; Dan Biderman, Data curation, Software, Formal analysis, Methodology; Daniel Birman, Resources, Software, Visualization; Niccolò Bonacchi, Conceptualization, Data curation, Software, Formal analysis, Methodology, Writing – review and editing; Sebastian A Bruijns, Conceptualization, Software, Formal analysis, Visualization, Methodology, Writing – original draft, Project administration, Writing – review and editing; Kelly Buchanan, Data curation, Methodology; Robert AA Campbell, Software, Visualization, Methodology; Matteo Carandini, Resources, Funding acquisition, Methodology; Gaelle A Chapuis, Resources, Data curation, Software, Supervision, Funding acquisition, Validation, Visualization, Methodology, Writing – original draft, Project administration, Writing – review and editing; Anne K Churchland, Conceptualization, Resources, Supervision, Funding acquisition, Validation, Methodology, Writing – original draft, Project administration, Writing – review and editing; M Felicia Davatolhagh, Data curation, Investigation, Writing – original draft, Project administration, Writing – review and editing; Hyun Dong Lee, Formal analysis, Visualization, Writing – original draft, Writing – review and editing; Mayo Faulkner, Data curation, Software, Visualization, Project administration; Berk Gerçek, Software, Validation, Methodology; Fei Hu, Anne E Urai, Conceptualization, Data curation, Investigation, Methodology; Julia Huntenburg, Data curation, Software; Cole Lincoln Hurwitz, Software, Formal analysis, Methodology; Anup Khanal, Christopher Krasniak, Petrina Lau, Data curation, Investigation, Methodology; Christopher Langfield, Software, Formal analysis, Validation; Nancy Mackenzie, Software, Validation, Visualization; Guido T Meijer, Conceptualization, Data curation, Software, Formal analysis, Validation, Investigation, Visualization, Methodology, Writing – original draft, Project administration, Writing – review and editing; Nathaniel J Miska, Investigation, Methodology; Zeinab Mohammadi, Conceptualization, Formal analysis, Validation, Methodology; Jean-Paul Noel, Conceptualization, Supervision, Investigation; Liam Paninski, Conceptualization, Data curation, Formal analysis, Supervision, Funding acquisition, Validation, Visualization, Methodology,

Writing – original draft, Project administration, Writing – review and editing; Alejandro Pan-Vazquez, Conceptualization, Data curation, Investigation; Cyrille Rossant, Software, Visualization; Noam Roth, Data curation, Software, Formal analysis, Supervision, Validation, Investigation, Visualization, Writing – original draft, Project administration, Writing – review and editing; Michael Schartner, Conceptualization, Resources, Data curation, Software, Formal analysis, Supervision, Validation, Visualization, Methodology, Writing – original draft, Project administration, Writing – review and editing; Karolina Z Socha, Conceptualization, Data curation, Software, Validation, Investigation, Methodology; Nicholas A Steinmetz, Conceptualization, Data curation, Supervision, Funding acquisition, Validation, Visualization, Methodology, Writing – original draft, Project administration, Writing – review and editing; Karel Svoboda, Conceptualization, Data curation, Formal analysis, Supervision, Funding acquisition, Validation, Methodology, Project administration, Writing – review and editing; Marsa Taheri, Conceptualization, Data curation, Software, Formal analysis, Supervision, Funding acquisition, Validation, Visualization, Methodology, Writing – original draft, Project administration, Writing – review and editing; Shuqi Wang, Software, Formal analysis, Visualization, Methodology, Writing – review and editing; Miles Wells, Conceptualization, Data curation, Project administration; Steven J West, Data curation, Software, Formal analysis, Supervision, Funding acquisition, Validation, Visualization, Methodology, Writing – original draft, Project administration, Writing – review and editing; Matthew R Whiteway, Data curation, Software, Formal analysis, Methodology, Writing – review and editing; Olivier Winter, Data curation, Software, Methodology, Writing – original draft; Ilana B Witten, Conceptualization, Resources, Supervision; Yizi Zhang, Software, Formal analysis, Visualization, Writing – review and editing

## Author ORCIDs
Daniel Birman ⓘ https://orcid.org/0000-0003-3748-6289
Matteo Carandini ⓘ https://orcid.org/0000-0003-4880-7682
Anne K Churchland ⓘ https://orcid.org/0000-0002-3205-3794
M Felicia Davatolhagh ⓘ https://orcid.org/0000-0002-0607-5164
Cole Lincoln Hurwitz ⓘ https://orcid.org/0000-0002-2023-1653
Jean-Paul Noel ⓘ https://orcid.org/0000-0001-5297-3363
Anne E Urai ⓘ https://orcid.org/0000-0001-5270-6513
Ilana B Witten ⓘ https://orcid.org/0000-0003-0548-2160
Yizi Zhang ⓘ https://orcid.org/0000-0001-6939-7868

## Ethics
All procedures and experiments were carried out in accordance with local laws and following approval by the relevant institutions: the Animal Welfare Ethical Review Body of University College London (Home Office approvals P1DB285D8, PCC4A4ECE, PD867676F); the Institutional Animal Care and Use Committees of Cold Spring Harbor Laboratory (approval ARC-2020-121), Princeton University (Approval 1876-20), University of California at Los Angeles (Approval ARC-2020-121-TR-001), and University of California at Berkeley; the University Animal Welfare Committee of New York University (Approval 1411117; 19.5); and the Portuguese Veterinary General Board (Approval 0421/0000/0000/2019).

Reviewer #1 (Public review): https://doi.org/10.7554/eLife.100840.3.sa1
Reviewer #2 (Public review): https://doi.org/10.7554/eLife.100840.3.sa2
Author response https://doi.org/10.7554/eLife.100840.3.sa3

# Additional files

## Supplementary files
MDAR checklist
Source data 1. Quality control spreadsheet.

## Data availability

All data are freely available. Please visit https://int-brain-lab.github.io/iblenv/notebooks_external/data_release_repro_ephys.html to access the data used in this article. Please visit the visualization website https://viz.internationalbrainlab.org/app to view the data (use the tab *Repeated sites*).

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
