## [Editor Report · eLife Assessment]

This paper represents an **important** contribution to the field. Summarizing results from neural recording experiments in mice across ten labs, the work provides **compelling** evidence that basic electrophysiology features, single-neuron functional properties, and population-level decoding are fairly reproducible across labs with proper preprocessing. The results and suggestions regarding preprocessing and quality metrics may be of significant interest to investigators carrying out such experiments in their own labs.

---

## [Referee Report · Reviewer #1 (Public review)]

The IBL here presents an important paper that aims to assess potential reproducibility issues in rodent electrophysiological recordings across labs and suggests solutions to these. The authors carried out a series of analyses on data collected across 10 laboratories while mice performed the same decision-making task, and provided convincing evidence that basic electrophysiology features, single-neuron functional properties, and population-level decoding were fairly reproducible across labs with proper preprocessing. This well-motivated large-scale collaboration allowed systematic assessment of lab-to-lab reproducibility of electrophysiological data, and the suggestions outlined in the paper for streamlining preprocessing pipelines and quality metrics will provide general guidance for the field, especially with continued effort to benchmark against standard practices (such as manual curation).

The authors have carefully incorporated our suggestions. As a result, the paper now better reflects where reproducibility is affected when using common, simple, and more complex analyses and preprocessing methods, and it is more informative-and more reflective of the field overall. We thank the reviewers for this thorough revision. We have 2 remaining suggestions on text clarification:

(1) Regarding benchmarking the automated metrics to manual curation of units: although we appreciate that a proper comparison may require a lot of effort potentially beyond the scope of the current paper; we do think that explicit discussion regarding this point is needed in the text, to remind the readers (and indeed future generations of electrophysiologists) the pros and cons of different approaches.

In addition to what the authors have currently stated (line 469-470):

"Another significant limitation of the analysis presented here is that we have not been able to assess the extent to which other choices of quality metrics and inclusion criteria might have led to greater or lesser reproducibility."

Maybe also add:

"In particular, a thorough comparison of automated metrics against a careful, large, manually-curated dataset, is an important benchmarking step for future studies.

(2) The authors now include in Figure 3-Figure Supplement 1 that highlight how much probe depth is adjusted by using electrophysiological features such as LFP power to estimate probe and channel depth. This plot is immensely informative for the field, as it implies that there can be substantial variability-sometimes up to 1 mm discrepancy between insertions-in depth estimation based on anatomical DiI track tips alone. Using electrophysiological features in this way for probe depth estimation is currently not standard in the field and has only been made possible with Neuropixels, which span several millimeters. These figures highlight that this should be a critical step in preprocessing pipelines, and the paper provides solid evidence for this.

Currently, this part of the figure is only subtly referenced to in the text. We think it would be helpful to explicitly reference this particular panel with discussions of its implication in the text.

---

## [Referee Report · Reviewer #2 (Public review)]

Summary:

The authors sought to evaluate whether analyses of large-scale electrophysiology data obtained from 10 different individual laboratories are reproducible when they use standardized procedures and quality control measures. They were able to reproduce most of their experimental findings across all labs. Despite attempting to target the same brain areas in each recording, variability in electrode targeting was a source of some differences between datasets.

Strengths:

This paper gathered a standardized dataset across 10 labs and performed a host of state-of-the-art analyses on it. Their ability to assess the reproducibility of each analysis across this kind of data is an important contribution to the field.

Comments on revisions:

The authors have addressed almost all of the concerns that I raised in this revised version. The new RIGOR notebook is helpful, as are the new analyses.

This paper attributes much error in probe insertion trajectory planning to the fact that the Allen CCF and standard stereotaxic coordinate systems are not aligned. Consequently, it would be very helpful for the community if this paper could recommend software tools, procedures, or code to do trajectory planning that accounts for this.

I think it would still be helpful for the paper to have some discussion comparing/contrasting the use of the RIGOR framework with existing spike sorting statistics. They mention in their response to reviewers that this is indeed a large space of existing approaches. Most labs performing Neuropixels recordings already do some type of quality control, but these approaches are not standardized. This work is well-positioned to discuss the advantages and disadvantages of these alternative approaches (even briefly) but does not currently do so-it does not need to run any of these competing approaches to helpfully mention ideas for what a reader of the paper should do for quality control with their own data.

---

## [Author Response]

The following is the authors’ response to the original reviews.

We thank the reviewers and editors for their careful read of our paper, and appreciate the thoughtful comments.

Both reviewers agreed that our work had several major strengths: the large dataset collected in collaboration across ten labs, the streamlined processing pipelines, the release of code repositories, the multi-task neural network, and that we definitively determined that electrode placement is an important source of variability between datasets.

However, a number of key potential improvements were noted: the reviewers felt that a more standard model-based characterization of single neuron responses would benefit our reproducibility analysis, that more detail was needed about the number of cells, sessions, and animals, and that more information was needed to allow users to deploy the RIGOR standards and to understand their relationship to other metrics in the field.

We agree with these suggestions and have implemented many major updates in our revised manuscript. Some highlights include:

(1) A new regression analysis that specifies the response profile of each neuron, allowing a comparison of how similar these are across labs and areas (See Figure 7 in the new section, “Single neuron coefficients from a regression-based analysis are rep oducible across labs”);

(2) A new decoding analysis (See Figure 9 in the section, “Decodability of task variables is consistent across labs, but varies by brain region”);

(3) A new RIGOR notebook to ease useability;

(4) A wealth of additional information about the cells, animals and sessions in each figure;

(5) Many new additional figure panels in the main text and supplementary material to clarify the specific points raised by the reviewers.

Again, we are grateful to the reviewers and editors for their helpful comments, which have significantly improved the work. We are hopeful that the many revisions we have implemented will be sufficient to change the “incomplete” designation that was originally assigned to the manuscript.

**Reviewer #1 (Public review):**
Summary:The authors explore a large-scale electrophysiological dataset collected in 10 labs while mice performed the same behavioral task, and aim to establish guidelines to aid reproducibility of results collected across labs. They introduce a series of metrics for quality control of electrophysiological data and show that histological verification of recording sites is important for interpreting findings across labs and should be reported in addition to planned coordinates. Furthermore, the authors suggest that although basic electrophysiology features were comparable across labs, task modulation of single neurons can be variable, particularly for some brain regions. The authors then use a multi-task neural network model to examine how neural dynamics relate to multiple interacting task- and experimenter-related variables, and find that lab-specific differences contribute little to the variance observed. Therefore, analysis approaches that account for correlated behavioral variables are important for establishing reproducible results when working with electrophysiological data from animals performing decision-making tasks. This paper is very well-motivated and needed. However, what is missing is a direct comparison of task modulation of neurons across labs using standard analysis practice in the fields, such as generalized linear model (GLM). This can potentially clarify how much behavioral variance contributes to the neural variance across labs; and more accurately estimate the scale of the issues of reproducibility in behavioral systems neuroscience, where conclusions often depend on these standard analysis methods.

We fully agree that a comparison of task-modulation across labs is essential. To address this, we have performed two new analyses and added new corresponding figures to the main text (Figures 7 and 9). As the reviewer hoped, this analysis did indeed clarify how much behavioral variance contributes to the variance across labs. Critically, these analyses suggested that our results were more robust to reproducibility than the more traditional analyses would indicate.

Additional details are provided below (See detailed response to R1P1b).

Strengths:(1) This is a well-motivated paper that addresses the critical question of reproducibility in behavioural systems neuroscience. The authors should be commended for their efforts.(2) A key strength of this study comes from the large dataset collected in collaboration across ten labs. This allows the authors to assess lab-to-lab reproducibility of electrophysiological data in mice performing the same decision-making task.(3) The authors' attempt to streamline preprocessing pipelines and quality metrics is highly relevant in a field that is collecting increasingly large-scale datasets where automation of these steps is increasingly needed.(4) Another major strength is the release of code repositories to streamline preprocessing pipelines across labs collecting electrophysiological data.(5) Finally, the application of MTNN for characterizing functional modulation of neurons, although not yet widely used in systems neuroscience, seems to have several advantages over traditional methods.

Thanks very much for noting these strengths of our work.

Weaknesses:(1) In several places the assumptions about standard practices in the field, including preprocessing and analyses of electrophysiology data, seem to be inaccurately presented:a) The estimation of how much the histologically verified recording location differs from the intended recording location is valuable information. Importantly, this paper provides citable evidence for why that is important. However, histological verification of recording sites is standard practice in the field, even if not all studies report them. Although we appreciate the authors' effort to further motivate this practice, the current description in the paper may give readers outside the field a false impression of the level of rigor in the field.

We agree that labs typically do perform histological verification. Still, our methods offer a substantial improvement over standard practice, and this was critical in allowing us to identify errors in targeting. For instance, we used new software, LASAGNA, which is an innovation over the traditional, more informal approach to localizing recording sites. Second, the requirement that two independent reviewers concur on each proposed location for a recording site is also an improvement over standard practice. Importantly, these reviewers use electrophysiological features to more precisely localize electrodes, when needed, which is an improvement over many labs. Finally, most labs use standard 2D atlases to identify recording location (a traditional approach); our use of a 3D atlas and a modern image registration pipeline has improved the accuracy of identifying the true placement of probes in 3D space.

Importantly, we don’t necessarily advocate that all labs adopt our pipeline; indeed, this would be infeasible for many labs. Instead, our hope is that the variability in probe trajectory that we uncovered will be taken into account in future studies. Here are 3 example ways in which that could happen. First, groups hoping to target a small area for an experiment might elect to use a larger cohort than previously planned, knowing that some insertions will miss their target. Second, our observation that some targeting error arose because experimenters had to move probes due to blood vessels will impact future surgeries: when an experimenter realizes that a blood vessel is in the way, they might still re-position the probe, but they can also adjust its trajectory (e.g., changing the angle) knowing that even little nudges to avoid blood vessels can have a large impact on the resulting insertion trajectory. Third, our observation of a 7 degree deviation between stereotaxic coordinates and Allen Institute coordinates can be used for future trajectory planning steps to improve accuracy of placement. Uncovering this deviation required many insertions and our standardized pipeline, but now that it is known, it can be easily corrected without needing such a pipeline.

We thank the reviewer for bringing up this issue and have added new text (and modified existing text) in the Discussion to highlight the innovations we introduced that allowed us to carefully quantify probe trajectory across labs (lines 500 - 515):

“Our ability to detect targeting error benefited from an automated histological pipeline combined with alignment and tracing that required agreement between multiple users, an approach that greatly exceeds the histological analyses done by most individual labs. Our approach, which enables scalability and standardization across labs while minimizing subjective variability, revealed that much of the variance in targeting was due to the probe entry positions at the brain surface, which were randomly displaced across the dataset. … Detecting this offset relied on a large cohort size and an automated histological pipeline, but now that we have identified the offset, it can be easily accounted for by any lab. Specifically, probe angles must be carefully computed from the CCF, as the CCF and stereotaxic coordinate systems do not define the same coronal plane angle. Minimizing variance in probe targeting is another important element in increasing reproducibility, as slight deviations in probe entry position and angle can lead to samples from different populations of neurons. Collecting structural MRI data in advance of implantation could reduce targeting error, although this is infeasible for most labs. A more feasible solution is to rely on stereotaxic coordinates but account for the inevitable off-target measurements by increasing cohort sizes and adjusting probe angles when blood vessels obscure the desired location.”

b) When identifying which and how neurons encode particular aspects of stimuli or behaviour in behaving animals (when variables are correlated by the nature of the animals behaviour), it has become the standard in behavioral systems neuroscience to use GLMs - indeed many labs participating in the IBL also has a long history of doing this (e.g., Steinmetz et al., 2019; Musall et al., 2023; Orsolic et al., 2021; Park et al., 2014). The reproducibility of results when using GLMs is never explicitly shown, but the supplementary figures to Figure 7 indicate that results may be reproducible across labs when using GLMs (as it has similar prediction performance to the MTNN). This should be introduced as the first analysis method used in a new dedicated figure (i.e., following Figure 3 and showing results of analyses similar to what was shown for the MTNN in Figure 7). This will help put into perspective the degree of reproducibility issues the field is facing when analyzing with appropriate and common methods. The authors can then go on to show how simpler approaches (currently in Figures 4 and 5) - not accounting for a lot of uncontrolled variabilities when working with behaving animals - may cause reproducibility issues.

We fully agree with the reviewer's suggestion. We have addressed their concern by implementing a Reduced-Rank Regression (RRR) model, which builds upon and extends the principles of Generalized Linear Models (GLMs). The RRR model retains the core regression framework of GLMs while introducing shared, *trainable* temporal bases across neurons, enhancing the model’s capacity to capture the structure in neural activity (Posani, Wang, et al., bioRxiv, 2024). Importantly, Posani, Wang et al compared the predictive performance of GLMs vs the RRR model, and found that the RRR model provided (slightly) improved performance, so we chose the RRR approach here.

We highlight this analysis in a new section (lines 350-377) titled, “Single neuron coefficients from a regression-based analysis are reproducible across labs”. This section includes an entirely new Figure (Fig. 7), where this new analysis felt most appropriate, since it is closer in spirit to the MTNN analysis that follows (rather than as a new Figure 3, as the reviewer suggested). As the reviewer hoped, this analysis provides some reassurance that including many variables when characterizing neural activity furnishes results with improved reproducibility. We now state this in the Results and the Discussion (line 456-457), highlighting that these analyses complement the more traditional selectivity analyses, and that using both methods together can be informative.

When the authors introduce a neural network approach (i.e. MTNN) as an alternative to the analyses in Figures 4 and 5, they suggest: 'generalized linear models (GLMs) are likely too inflexible to capture the nonlinear contributions that many of these variables, including lab identity and spatial positions of neurons, might make to neural activity'. This is despite the comparison between MTNN and GLM prediction performance (Supplement 1 to Figure 7) showing that the MTNN is only slightly better at predicting neural activity compared to standard GLMs. The introduction of new models to capture neural variability is always welcome, but the conclusion that standard analyses in the field are not reproducible can be unfair unless directly compared to GLMs.In essence, it is really useful to demonstrate how different analysis methods and preprocessing approaches affect reproducibility. But the authors should highlight what is actually standard in the field, and then provide suggestions to improve from there.

Thanks again for these comments. We have also edited the MTNN section slightly to accommodate the addition of the previous new RRR section (line 401-402).

(2) The authors attempt to establish a series of new quality control metrics for the inclusion of recordings and single units. This is much needed, with the goal to standardize unit inclusion across labs that bypasses the manual process while keeping the nuances from manual curation. However, the authors should benchmark these metrics to other automated metrics and to manual curation, which is still a gold standard in the field. The authors did this for whole-session assessment but not for individual clusters. If the authors can find metrics that capture agreed-upon manual cluster labels, without the need for manual intervention, that would be extremely helpful for the field.

We thank the reviewer for their insightful suggestions regarding benchmarking our quality control metrics against manual curation and other automated methods at the level of individual clusters. We are indeed, as the reviewer notes, publishing results from spike sorting outputs that have been automatically but not manually verified on a neuron-by-neuron basis. To get to the point where we trust these results to be of publishable quality, we manually reviewed hundreds of recordings and thousands of neurons, refining both the preprocessing pipeline and the single-unit quality metrics along the way. All clusters, both those passing QCs and those not passing QCs, are available to review with detailed plots and quantifications at https://viz.internationalbrainlab.org/app (turn on “show advanced metrics” in the upper right, and navigate to the plots furthest down the page, which are at the individual unit level). We would emphasize that these metrics are definitely imperfect (and fully-automated spike sorting remains a work in progress), but so is manual clustering. Our fully automated approach has the advantage of being fully reproducible, which is absolutely critical for the analyses in the present paper. Indeed, if we had actually done manual clustering or curation, one would wonder whether our results were actually reproducible independently. Nevertheless, it is not part of the present manuscript’s objectives to validate or defend these specific choices for automated metrics, which have been described in detail elsewhere (see our Spike Sorting whitepaper, https://figshare.com/articles/online_resource/Spike_sorting_pipeline_for_the_International_Brain_Laboratory/19705522?file=49783080). It would be a valuable exercise to thoroughly compare these metrics against a careful, large, manually-curated set, but doing this properly would be a paper in itself and is beyond the scope of the current paper. We also acknowledge that our analyses studying reproducibility across labs could, in principle, result in more or less reproducibility under a different choice of metrics, which we now describe in the Discussion (line 469-470)”:

“Another significant limitation of the analysis presented here is that we have not been able to assess the extent to which other choices of quality metrics and inclusion criteria might have led to greater or lesser reproducibility.”

(3) With the goal of improving reproducibility and providing new guidelines for standard practice for data analysis, the authors should report of n of cells, sessions, and animals used in plots and analyses throughout the paper to aid both understanding of the variability in the plots - but also to set a good example.

We wholeheartedly agree and have added the number of cells, mice and sessions for each figure. This information is included as new tabs in our quality control spreadsheet (https://docs.google.com/spreadsheets/d/1_bJLDG0HNLFx3SOb4GxLxL52H4R2uPRcpUlIw6n4n-E/). This is referred to in line 158-159 (as well as its original location on line 554 in the section, “Quality control and data inclusion”).

Other general comments:(1) In the discussion (line 383) the authors conclude: 'This is reassuring, but points to the need for large sample sizes of neurons to overcome the inherent variability of single neuron recording'. - Based on what is presented in this paper we would rather say that their results suggest that appropriate analytical choices are needed to ensure reproducibility, rather than large datasets - and they need to show whether using standard GLMs actually allows for reproducible results.

Thanks. The new GLM-style RRR analysis in Figure 7, following the reviewer’s suggestion, does indeed indicate improved reproducibility across labs. As described above, we see this new analysis as complementary to more traditional analyses of neural selectivity and argue that the two can be used together. The new text (line 461) states:

“This is reassuring, and points to the need for appropriate analytical choices to ensure reproducibility.”

(2) A general assumption in the across-lab reproducibility questions in the paper relies on intralab variability vs across-lab variability. An alternative measure that may better reflect experimental noise is across-researcher variability, as well as the amount of experimenter experience (if the latter is a factor, it could suggest researchers may need more training before collecting data for publication). The authors state in the discussion that this is not possible. But maybe certain measures can be used to assess this (e.g. years of conducting surgeries/ephys recordings etc)?

We agree that understanding experimenter-to-experimenter variability would be very interesting and indeed we had hoped to do this analysis for some time. The problem is that typically, each lab employed one trainee to conduct all the data collection. This prevents us from comparing outcomes from two different experimenters in the same lab. There are exceptions to this, such as the Churchland lab in which 3 personnel (two postdocs and a technician) collected the data. However, even this fortuitous situation did not lend itself well to assessing experimenter-to-experimenter variation: the Churchland lab moved from Cold Spring Harbor to UCLA during the data collection period, which might have caused variability that is totally independent of experimenter (e.g., different animal facilities). Further, once at UCLA, the postdoc and technician worked closely together- alternating roles in animal training, surgery and electrophysiology. We believe that the text in our current Discussion (line 465-468) accurately characterizes the situation:

“Our experimental design precludes an analysis of whether the reproducibility we observed was driven by person-to-person standardization or lab-to-lab standardization. Most likely, both factors contributed: all lab personnel received standardized instructions for how to implant head bars and train animals, which likely reduced personnel-driven differences.”

Quantifying the level of experience of each experimenter is an appealing idea and we share the reviewer’s curiosity about its impact on data quality. Unfortunately, quantifying experience is tricky. For instance, years of conducting surgeries is not an unambiguously determinable number. Would we count an experimenter who did surgery every day for a year as having the same experience as an experimenter who did surgery once/month for a year? Would we count a surgeon with expertise in other areas (e.g., windows for imaging) in the same way as surgeons with expertise in ephys-specific surgeries? Because of the ambiguities, we leave this analysis to be the subject of future work; this is now stated in the Discussion (line 476).

(3) Figure 3b and c: Are these plots before or after the probe depth has been adjusted based on physiological features such as the LFP power? In other words, is the IBL electrophysiological alignment toolbox used here and is the reliability of location before using physiological criteria or after? Beyond clarification, showing both before and after would help the readers to understand how much the additional alignment based on electrophysiological features adjusts probe location. It would also be informative if they sorted these penetrations by which penetrations were closest to the planned trajectory after histological verification.

The plots in Figure 3b and 3c reflect data *after* the probe depth has been adjusted based on electrophysiological features. This adjustment incorporates criteria such as LFP power and spiking activity to refine the trajectory and ensure precise alignment with anatomical landmarks. The trajectories have also been reviewed and confirmed by two independent reviewers. We have clarified this in line 180 and in the caption of Figure 3.

To address this concern, we have added a new panel c in Figure 3 supplementary 1 (also shown below) that shows the LFP features along the probes prior to using the IBL alignment toolbox. We hope the reviewer agrees that a comparison of panels (a) and (c) below make clear the improvement afforded by our alignment tools.

In Figure 3 and Figure 3 supplementary 1, as suggested, we have also now sorted the probes by those that were closest to the planned trajectory. This way of visualizing the data makes it clear that as the distance from the planned trajectory increases, the power spectral density in the hippocampal regions becomes less pronounced and the number of probes that have a large portion of the channels localized to VISa/am, LP and PO decreases. We have added text to the caption to describe this. We thank the reviewer for this suggestion and agree that it will help readers to understand how much the additional alignment (based on electrophysiological features) adjusts probe location.

(4) In Figures 4 and 6: If the authors use a 0.05 threshold (alpha) and a cell simply has to be significant on 1/6 tests to be considered task modulated, that means that they have a false positive rate of ~30% (0.05*6=0.3). We ran a simple simulation looking for significant units (from random null distribution) from these criteria which shows that out of 100.000 units, 26500 units would come out significant (false error rate: 26.5%). That is very high (and unlikely to be accepted in most papers), and therefore not surprising that the fraction of task-modulated units across labs is highly variable. This high false error rate may also have implications for the investigation of the spatial position of task-modulated units (as effects of the spatial position may drown in falsely labelled 'task-modulated' cells).

Thank you for this concern. The different tests were kept separate, so we did not consider a neuron modulated if it was significant in only one out of six tests, but instead we asked whether a neuron was modulated according to test one, whether it was modulated according to test two, etc., and performed further analyses separately for each test. Thus, we are only vulnerable to the ‘typical’ false positive rate of 0.05 for any given test. We made this clearer in the text (lines 232-236) and hope that the 5% false positive rate seems more acceptable.

(5) The authors state from Figure 5b that the majority of cells could be well described by 2 PCs. The distribution of R2 across neurons is almost uniform, so depending on what R2 value one considers a 'good' description, that is the fraction of 'good' cells. Furthermore, movement onset has now been well-established to be affecting cells widely and in large fractions, so while this analysis may work for something with global influence - like movement - more sparsely encoded variables (as many are in the brain) may not be well approximated with this suggestion. The authors could expand this analysis into other epochs like activity around stimulus presentation, to better understand how this type of analysis reproduces across labs for features that have a less global influence.

We thank the reviewer for the suggestion and fully agree that the window used in our original analysis would tend to favor movement-driven neurons. To address this, we repeated the analysis, this time using a window centered around stimulus onset (from -0.5 s prior to stimulus onset until 0.1 s after stimulus onset). As the reviewer suspected, far fewer neurons were active in this window and consequently far fewer were modelled well by the first two PCs, as shown in Author response image 1b (below). Similar to our original analysis using the post-movement window, we found mixed results for the stimulus-centered window across labs. Interestingly, regional differences were weaker in this new analysis compared to the original analysis of the post-movement window. We have added a sentence to the results describing this. Because the results are similar to the post-movement window main figure, we would prefer to restrict the new analysis only to this point-by-point response, in the hopes of streamlining the paper.

**Author response image 1. sa3fig1:** PCA analysis applied to a stimulus-aligned window ([-0.5, 0.1] sec relative to stim onset). Figure conventions as in main text Fig 5. Results are comparable to the post-movement window analysis, however regional differences are weaker here, possibly because fewer cells were active in the pre-movement window. We added panel j here and in the main figure, showing cell-number-controlled results. I.e. for each test, the minimum neuron number of the compared classes was sampled from all classes (say labs in a region), this sampling was repeated 1000 times and p-values combined via Fisher’s method, overall resulting in much fewer significant differences across laboratories and, independently, regions.

(6) Additionally, in Figure 5i: could the finding that one can only distinguish labs when taking cells from all regions, simply be a result of a different number of cells recorded in each region for each lab? It makes more sense to focus on the lab/area pairing as the authors also do, but not to make their main conclusion from it. If the authors wish to do the comparison across regions, they will need to correct for the number of cells recorded in each region for each lab. In general, it was a struggle to fully understand the purpose of Figure 5. While population analysis and dimensionality reduction are commonplace, this seems to be a very unusual use of it.

We agree that controlling for varying cell numbers is a valuable addition to this analysis. We added panel j in Fig. 5 showing cell-number-controlled test results of panel i. I.e. for a given statistical comparison, we sample the lowest number of cells of compared classes from the others, do the test, and repeat this sampling 1000 times, before combining the p-values using Fisher’s method. This cell-number controlled version of the tests resulted in clearly fewer significant differences across distributions - seen similarly for the pre-movement window shown in j in Author response image 1. We hope this clarified our aim to illustrate that low-dimensional embedding of cells’ trial-averaged activity can show how regional differences compare with laboratory differences.

As a complementary statistical analysis to the shown KS tests, we fitted a linear-mixed-effects model (statsmodels.formula.api mixedlm), to the first and second PC for both activity windows (“Move”: [-0.5,1] first movement aligned; “Stim”: [-0.5,0.1] stimulus onset aligned), independently. Author response image 2 (in this rebuttal only) is broadly in line with the KS results, showing more regional than lab influences on the distributions of first PCs for the post-movement window.

**Author response image 2. sa3fig2:** Linear mixed effects model results for two PCs and two activity windows. For the post-movement window (“Move”), regional influences are significant (red color in plots) for all but one region while only one lab has a significant model coefficient for PC1. For PC2 more labs and three regions have significant coefficients. For the pre-movement window (“Stim”) one region for PC1 or PC2 has significant coefficients. The variance due to session id was smaller than all other effects (“eids Var”). “Intercept” shows the expected value of the response variable (PC1, PC2) before accounting for any fixed or random effects. All p-values were grouped as one hypothesis family and corrected for multiple comparisons via Benjamini-Hochberg.

(7) In the discussion the authors state: " Indeed this approach is a more effective and streamlined way of doing it, but it is questionable whether it 'exceeds' what is done in many labs.Classically, scientists trace each probe manually with light microscopy and designate each area based on anatomical landmarks identified with nissl or dapi stains together with gross landmarks. When not automated with 2-PI serial tomography and anatomically aligned to a standard atlas, this is a less effective process, but it is not clear that it is less precise, especially in studies before neuropixels where active electrodes were located in a much smaller area. While more effective, transforming into a common atlas does make additional assumptions about warping the brain into the standard atlas - especially in cases where the brain has been damaged/lesioned. Readers can appreciate the effectiveness and streamlining provided by these new tools without the need to invalidate previous approaches.

We thank the reviewer for highlighting the effectiveness of manual tracing methods used traditionally. Our intention in the statement was not to invalidate the precision or value of these classical methods but rather to emphasize the scalability and streamlining offered by our pipeline. We have revised the language to more accurately reflect this (line 500-504):

“Our ability to detect targeting error benefited from an automated histological pipeline combined with alignment and tracing that required agreement between multiple users, an approach that greatly exceeds the histological analyses done by most individual labs. Our approach, which enables scalability and standardization across labs while minimizing subjective variability, revealed that much of the variance in targeting was due to the probe entry positions at the brain surface, which were randomly displaced across the dataset.”

(8) What about across-lab population-level representation of task variables, such as in the coding direction for stimulus or choice? Is the general decodability of task variables from the population comparable across labs?

Excellent question, thanks! We have added the new section “Decodability of task variables is consistent across labs, but varies by brain region” (line 423-448) and Figure 9 in the revised manuscript to address this question. In short, yes, the general decodability of task variables from the population is comparable across labs, providing additional reassurance of reproducibility.

**Reviewer #2 (Public review):**
Summary:The authors sought to evaluate whether observations made in separate individual laboratories are reproducible when they use standardized procedures and quality control measures. This is a key question for the field. If ten systems neuroscience labs try very hard to do the exact same experiment and analyses, do they get the same core results? If the answer is no, this is very bad news for everyone else! Fortunately, they were able to reproduce most of their experimental findings across all labs. Despite attempting to target the same brain areas in each recording, variability in electrode targeting was a source of some differences between datasets.Major Comments:The paper had two principal goals:(1) to assess reproducibility between labs on a carefully coordinated experiment(2) distill the knowledge learned into a set of standards that can be applied across the field.The manuscript made progress towards both of these goals but leaves room for improvement.(1) The first goal of the study was to perform exactly the same experiment and analyses across 10 different labs and see if you got the same results. The rationale for doing this was to test how reproducible large-scale rodent systems neuroscience experiments really are. In this, the study did a great job showing that when a consortium of labs went to great lengths to do everything the same, even decoding algorithms could not discern laboratory identity was not clearly from looking at the raw data. However, the amount of coordination between the labs was so great that these findings are hard to generalize to the situation where similar (or conflicting!) results are generated by two labs working independently.Importantly, the study found that electrode placement (and thus likely also errors inherent to the electrode placement reconstruction pipeline) was a key source of variability between datasets. To remedy this, they implemented a very sophisticated electrode reconstruction pipeline (involving two-photon tomography and multiple blinded data validators) in just one lab-and all brains were sliced and reconstructed in this one location. This is a fantastic approach for ensuring similar results within the IBL collaboration, but makes it unclear how much variance would have been observed if each lab had attempted to reconstruct their probe trajectories themselves using a mix of histology techniques from conventional brain slicing, to light sheet microscopy, to MRI imaging.This approach also raises a few questions. The use of standard procedures, pipelines, etc. is a great goal, but most labs are trying to do something unique with their setup. Bigger picture, shouldn't highly "significant" biological findings akin to the discovery of place cells or grid cells, be so clear and robust that they can be identified with different recording modalities and analysis pipelines?

We agree, and hope that this work may help readers understand what effect sizes may be considered “clear and robust” from datasets like these. We certainly support the reviewer’s point that multiple approaches and modalities can help to confirm any biological findings, but we would contend that a clear understanding of the capabilities and limitations of each approach is valuable, and we hope that our paper helps to achieve this.

Related to this, how many labs outside of the IBL collaboration have implemented the IBL pipeline for their own purposes? In what aspects do these other labs find it challenging to reproduce the approaches presented in the paper? If labs were supposed to perform this same experiment, but without coordinating directly, how much more variance between labs would have been seen? Obviously investigating these topics is beyond the scope of this paper. The current manuscript is well-written and clear as is, and I think it is a valuable contribution to the field. However, some additional discussion of these issues would be helpful.

We thank the reviewer for raising this important issue. We know of at least 13 labs that have implemented the behavioral task software and hardware that we published in eLife in 2021, and we expect that over the next several years labs will also implement these analysis pipelines (note that it is considerably cheaper and faster to implement software pipelines than hardware). In particular, a major goal of the staff in the coming years is to continue and improve the support for pipeline deployment and use. However, our goal in this work, which we have aimed to state more clearly in the revised manuscript, was not so much to advocate that others adopt our pipeline, but instead to use our standardized approach as a means of assessing reproducibility under the best of circumstances (see lines 48-52): “A high level of reproducibility of results across laboratories when procedures are carefully matched is a prerequisite to reproducibility in the more common scenario in which two investigators approach the same high-level question with slightly different experimental protocols.”

Further, a number of our findings are relevant to other labs regardless of whether they implement our exact pipeline, a modified version of our pipeline, or something else entirely. For example, we found probe targeting to be a large source of variability. Our ability to detect targeting error benefited from an automated histological pipeline combined with alignment and tracing that required agreement between multiple users, but now that we have identified the offset, it can be easily accounted for by any lab. Specifically, probe angles must be carefully computed from the CCF, as the CCF and stereotaxic coordinate systems do not define the same coronal plane angle. Relatedly, we found that slight deviations in probe entry position can lead to samples from different populations of neurons. Although this took large cohort sizes to discover, knowledge of this discovery means that future experiments can plan for larger cohort sizes to allow for off-target trajectories, and can re-compute probe angle when the presence of blood vessels necessitates moving probes slightly. These points are now highlighted in the Discussion (lines 500-515).

Second, the proportion of responsive neurons (a quantity often used to determine that a particular area subserves a particular function), sometimes failed to reproduce across labs. For example, for movement-driven activity in PO, UCLA reported an average change of 0 spikes/s, while CCU reported a large and consistent change (Figure 4d, right most panel, compare orange vs. yellow traces). This argues that neuron-to-neuron variability means that comparisons across labs require large cohort sizes. A small number of outlier neurons in a session can heavily bias responses. We anticipate that this problem will be remedied as tools for large scale neural recordings become more widely used. Indeed, the use of 4-shank instead of single-shank Neuropixels (as we used here) would have greatly enhanced the number of PO neurons we measured in each session. We have added new text to Results explaining this (lines 264-268):

“We anticipate that the feasibility of even larger scale recordings will make lab-to-lab comparisons easier in future experiments; multi-shank probes could be especially beneficial for cortical recordings, which tend to be the most vulnerable to low cell counts since the cortex is thin and is the most superficial structure in the brain and thus the most vulnerable to damage. Analyses that characterize responses to multiple parameters are another possible solution (See Figure 7).”

(2) The second goal of the study was to present a set of data curation standards (RIGOR) that could be applied widely across the field. This is a great idea, but its implementation needs to be improved if adoption outside of the IBL is to be expected. Here are three issues:(a) The GitHub repo for this project (https://github.com/int-brain-lab/paper-reproducible-ephys/) is nicely documented if the reader's goal is to reproduce the figures in the manuscript. Consequently, the code for producing the RIGOR statistics seems mostly designed for re-computing statistics on the existing IBL-formatted datasets. There doesn't appear to be any clear documentation about how to run it on arbitrary outputs from a spike sorter (i.e. the inputs to Phy).

We agree that clear documentation is key for others to adopt our standards. To address this, we have added a section at the end of the README of the repository that links to a jupyter notebook (https://github.com/int-brain-lab/paper-reproducible-ephys/blob/master/RIGOR_script.ipynb) that runs the RIGOR metrics on a user’s own spike sorted dataset. The notebook also contains a tutorial that walks through how to visually assess the quality of the raw and spike sorted data, and computes the noise level metrics on the raw data as well as the single cell metrics on the spike sorted data.

(b) Other sets of spike sorting metrics that are more easily computed for labs that are not using the IBL pipeline already exist (e.g. "quality_metrics" from the Allen Institute ecephys pipeline [https://github.com/AllenInstitute/ecephys_spike_sorting/blob/main/ecephys_spike_sorting/modules/quality_metrics/README.md] and the similar module in the Spike Interface package [https://spikeinterface.readthedocs.io/en/latest/modules/qualitymetrics.html]). The manuscript does not compare these approaches to those proposed here, but some of the same statistics already exist (amplitude cutoff, median spike amplitude, refractory period violation).

There is a long history of researchers providing analysis algorithms and code for spike sorting quality metrics, and we agree that the Allen Institute’s ecephys code and the Spike Interface package are the current options most widely used (but see also, for example, Fabre et al. https://github.com/Julie-Fabre/bombcell). Our primary goal in the present work is not to advocate for a particular implementation of any quality metrics (or any spike sorting algorithm, for that matter), but instead to assess reproducibility of results, given one specific choice of spike sorting algorithm and quality metrics. That is why, in our comparison of yield across datasets (Fig 1F), we downloaded the raw data from those comparison datasets and re-ran them under our single fixed pipeline, to establish a fair standard of comparison. A full comparison of the analyses presented here under different choices of quality metrics and spike sorting algorithms would undoubtedly be interesting and useful for the field - however, we consider it to be beyond the scope of the present work. It is therefore an important assumption of our work that the result would not differ materially under a different choice of sorting algorithm and quality metrics. We have added text to the Discussion to clarify this limitation:

“Another significant limitation of the analysis presented here is that we have not been able to assess the extent to which other choices of quality metrics and inclusion criteria might have led to greater or lesser reproducibility.”

That said, we still intend for external users to be able to easily run our pipelines and quality metrics.

(c) Some of the RIGOR criteria are qualitative and must be visually assessed manually. Conceptually, these features make sense to include as metrics to examine, but would ideally be applied in a standardized way across the field. The manuscript doesn't appear to contain a detailed protocol for how to assess these features. A procedure for how to apply these criteria for curating non-IBL data (or for implementing an automated classifier) would be helpful.

We agree. To address this, we have provided a notebook that runs the RIGOR metrics on a user’s own dataset, and contains a tutorial on how to interpret the resulting plots and metrics (https://github.com/int-brain-lab/paper-reproducible-ephys/blob/master/RIGOR_script.ipynb).

Within this notebook there is a section focused on visually assessing the quality of both the raw data and the spike sorted data. The code in this section can be used to generate plots, such as raw data snippets or the raster map of the spiking activity, which are typically used to visually assess the quality of the data. In Figure 1 Supplement 2 we have provided examples of such plots that show different types of artifactual activity that should be inspected.

Other Comments:(1) How did the authors select the metrics they would use to evaluate reproducibility? Was this selection made before doing the study?

Our metrics were selected on the basis of our experience and expertise with extracellular electrophysiology. For example: some of us previously published on epileptiform activity and its characteristics in some mice (Steinmetz et al. 2017), so we included detection of that type of artifact here; and, some of us previously published detailed investigations of instability in extracellular electrophysiological recordings and methods for correcting them (Steinmetz et al. 2021, Windolf et al. 2024), so we included assessment of that property here. These metrics therefore represent our best expert knowledge about the kinds of quality issues that can affect this type of dataset, but it is certainly possible that future investigators will discover and characterize other quality issues.

The selection of metrics was primarily performed before the study (we used these assessments internally before embarking on the extensive quantifications reported here), and in cases where we refined them further during the course of preparing this work, it was done without reference to statistical results on reproducibility but instead on the basis of manual inspection of data quality and metric performance.

(2) Was reproducibility within-lab dependent on experimenter identity?

We thank the reviewer for this question. We have addressed it in our response to R1 General comment 2, as follows:

We agree that understanding experimenter-to-experimenter variability would be very interesting and indeed we had hoped to do this analysis for some time. The problem is that typically, each lab employed one trainee to conduct all the data collection. This prevents us from comparing outcomes from two different experimenters in the same lab. There are exceptions to this, such as the Churchland lab in which 3 personnel (two postdocs and a technician) collected the data. However, even this fortuitous situation did not lend itself well to assessing experimenter-to-experimenter variation: the Churchland lab moved from Cold Spring Harbor to UCLA during the data collection period, which might have caused variability that is totally independent of experimenter (e.g., different animal facilities). Further, once at UCLA, the postdoc and technician worked closely together- alternating roles in animal training, surgery and electrophysiology. We believe that the text in our current Discussion (line 465-468) accurately characterizes the situation:

“Our experimental design precludes an analysis of whether the reproducibility we observed was driven by person-to-person standardization or lab-to-lab standardization. Most likely, both factors contributed: all lab personnel received standardized instructions for how to implant head bars and train animals, which likely reduced personnel-driven differences.”

Quantifying the level of experience of each experimenter is an appealing idea and we share the reviewer’s curiosity about its impact on data quality. Unfortunately, quantifying experience is tricky. For instance, years of conducting surgeries is not an unambiguously determinable number. Would we count an experimenter who did surgery every day for a year as having the same experience as an experimenter who did surgery once/month for a year? Would we count a surgeon with expertise in other areas (e.g., windows for imaging) in the same way as surgeons with expertise in ephys-specific surgeries? Because of the ambiguities, we leave this analysis to be the subject of future work; this is now stated in the Discussion (line 476).

(3) They note that UCLA and UW datasets tended to miss deeper brain region targets (lines 185-188) - they do not speculate why these labs show systematic differences. Were they not following standardized procedures?

Thank you for raising this point. All researchers across labs were indeed following standardised procedures. We note that our statistical analysis of probe targeting coordinates and angles did not reveal a significant effect of lab identity on targeting error, even though we noted the large number of mis-targeted recordings in UCLA and UW to help draw attention to the appropriate feature in the figure. Given that these differences were not statistically significant, we can see how it was misleading to call out these two labs specifically. While the overall probe placement surface error and angle error both show no such systematic difference, the magnitude of surface error showed a non-significant tendency to be higher for samples in UCLA & UW, which, compounded with the direction of probe angle error, caused these probe insertions to land in a final location outside LP & PO.

This shows how subtle differences in probe placement & angle accuracy can lead to compounded inaccuracies at the probe tip, especially when targeting deep brain regions, even when following standard procedures. We believe this is driven partly by the accuracy limit or resolution of the stereotaxic system, along with slight deviations in probe angle, occurring during the setup of the stereotaxic coordinate system during these recordings.

We have updated the relevant text in lines 187-190 as follows, to clarify:

“Several trajectories missed their targets in deeper brain regions (LP, PO), as indicated by gray blocks, despite the lack of significant lab-dependent effects in targeting as reported above. These off-target trajectories tended to have both a large displacement from the target insertion coordinates and a probe angle that unfavorably drew the insertions away from thalamic nuclei (Figure 2f).”

(4) The authors suggest that geometrical variance (difference between planned and final identified probe position acquired from reconstructed histology) in probe placement at the brain surface is driven by inaccuracies in defining the stereotaxic coordinate system, including discrepancies between skull landmarks and the underlying brain structures. In this case, the use of skull landmarks (e.g. bregma) to determine locations of brain structures might be unreliable and provide an error of ~360 microns. While it is known that there is indeed variance in the position between skull landmarks and brain areas in different animals, the quantification of this error is a useful value for the field.

We thank the reviewer for their thoughtful comment and are glad that they found the quantification of variance useful for the field.

(5) Why are the thalamic recording results particularly hard to reproduce? Does the anatomy of the thalamus simply make it more sensitive to small errors in probe positioning relative to the other recorded areas?

We thank the reviewer for raising this interesting question. We believe that they are referring to Figure 4: indeed when we analyzed the distribution of firing rate modulations, we saw some failures of reproducibility in area PO (bottom panel, Figure 4h). However, the thalamic nuclei were not, in other analyses, more vulnerable to failures in reproducibility. For example, in the top panel of Figure 4h, VisAM shows failures of reproducibility for modulation by the visual stimulus. In Fig. 5i, area CA1 showed a failure of reproducibility. We fear that the figure legend title in the previous version (which referred to the thalamus specifically) was misleading, and we have revised this. The new title is, “Neural activity is modulated during decision-making in five neural structures and is variable between laboratories.” This new text more accurately reflects that there were a number of small, idiosyncratic failures of reproducibility, but that these were not restricted to a specific structure. The new analysis requested by R1 (now in Figure 7) provides further reassurance of overall reproducibility, including in the thalamus (see Fig. 7a, right panels; lab identity could not be decoded from single neuron metrics, even in the thalamus).

**Reviewer #1 (Recommendations for the authors):**
(1) Figure font sizes and formatting are variable across panels and figures. Please streamline the presentation of results.

Thank you for your feedback. We have remade all figures with the same standardized font sizes and formatting.

(2) Please correct the noncontinuous color scales in Figures 3b and 3d.

Thank you for pointing this out, we fixed the color bar.

(3) In Figures 5d and g, the error bars are described as: 'Error bands are standard deviation across cells normalised by the square root of the number of sessions in the region'. How does one interpret this error? It seems to be related to the standard error of the mean (std/sqrt(n)) but instead of using the n from which the standard deviation is calculated (in this case across cells), the authors use the number of sessions as n. If they took the standard deviation across sessions this would be the sem across sessions, and interpretable (as sem*1.96 is the 95% parametric confidence interval of the mean). Please justify why these error bands are used here and how they can be interpreted - it also seems like it is the only time these types of error bands are used.

We agree and for clarity use standard error across cells now, as the error bars do not change dramatically either way.

(4) It is difficult to understand what is plotted in Figures 5e,h, please unpack this further and clarify.

Thank you for pointing this out. We have added additional explanation in the figure caption (See caption for Figure 5c) to explain the KS test.

(5) In lines 198-201 the authors state that they were worried that Bonferroni correction with 5 criteria would be too lenient, and therefore used 0.01 as alpha. I am unsure whether the authors mean that they are correcting for multiple comparisons across features or areas. Either way, 0.01 alpha is exactly what a Bonferroni corrected alpha would be when correcting for either 5 features or 5 areas: 0.05/5=0.01. Or do they mean they apply the Bonferroni correction to the new 0.01 alpha: i.e., 0.01/5=0.002? Please clarify.

Thank you, that was indeed written confusingly. We considered all tests and regions as whole, so 7 tests * 5 regions = 35 tests, which would result in a very strong Bonferroni correction. Indeed, if one considers the different tests individually, the correction we apply from 0.05 to 0.01 can be considered as correcting for the number of regions, which we now highlight better. We apply no further corrections of any kind to our alpha=0.01. We clarified this in the manuscript in all relevant places (lines 205-208, 246, 297-298, and 726-727).

(6) Did the authors take into account how many times a probe was used/how clean the probe was before each recording. Was this streamlined between labs? This can have an effect on yield and quality of recording.

We appreciate the reviewer highlighting the potential impact of probe use and cleanliness on recording quality and yield. While we did not track the number of times each probe was used, we ensured that all probes were cleaned thoroughly after each use using a standardized cleaning protocol (Section 16: Cleaning the electrode after data acquisition in Appendix 2: IBL protocol for electrophysiology recording using Neuropixels probe). We acknowledge that tracking the specific usage history of each probe could provide additional insights, but unfortunately we did not track this information for this project. In prior work the re-usability of probes has been quantified, showing insignificant degradation with use (e.g. Extended Data Fig 7d from Jun et al. 2017).

(7) Figure 3, Supplement1: DY_013 missed DG entirely? Was this included in the analysis?

Thank you for this question. We believe the reviewer is referring to the lack of a prominent high-amplitude LFP band in this mouse, and lack of high-quality sorted units in that region. Despite this, our histology did localize the recording trajectory to DG. This recording did pass our quality control criteria overall, as indicated by the green label, and was used in relevant analyses.

The lack of normal LFP features and neuron yield might reflect the range of biological variability (several other sessions also have relatively weak DG LFP and yield, though DY_013 is the weakest), or could reflect some damage to the tissue, for example as caused by local bleeding. Because we could not conclusively identify the source of this observation, we did not exclude it.

(8) Given that the authors argue for using the MTNN over GLMs, it would be useful to know exactly how much better the MTNN is at predicting activity in the held-out dataset (shown in Figure 7, Supplement 1). It looks like a very small increase in prediction performance between MTNN and GLMs, is it significantly different?

The average variance explained on the held-out dataset, as shown in Figure 8–Figure Supplement 1 Panel B, is 0.065 for the GLMs and 0.071 for the MTNN. As the reviewer correctly noted, this difference is not significant. However, one of the key advantages of the MTNN over GLMs lies in its flexibility to easily incorporate covariates, such as electrophysiological characteristics or session/lab IDs, directly into the analysis. This feature is particularly valuable for assessing effect sizes and understanding the contributions of various factors.

(9) In line 723: why is the threshold for mean firing rate for a unit to be included in the MTNN results so high (>5Hz), and how does it perform on units with lower firing rates?

We thank the reviewer for pointing this out. The threshold for including units with a mean firing rate above 5 Hz was set because most units with firing rates below this threshold were silent in many trials, and reducing the number of units helped keep the MTNN training time reasonable. Based on this comment, we ran the MTNN experiments including all units with firing rates above 1 Hz, and the results remained consistent with our previous conclusions (Figure 8). Crucially, the leave-one-out analysis consistently showed that lab and session IDs had effect sizes close to zero, indicating that both within-lab and between-lab random effects are small and comparable.

**Reviewer #2 (Recommendations for the authors):**
(1) Most of the more major issues were already listed in the above comments. The strongest recommendation for additional work would be to improve the description and implementation of the RIGOR statistics such that non-IBL labs that might use Neuropixels probes but not use the entire IBL pipeline might be able to apply the RIGOR framework to their own data.

We thank the reviewer for highlighting the importance of making the RIGOR statistics more accessible to a broader audience. We agree that improving the description and implementation of the RIGOR framework is essential for facilitation of non-IBL labs using Neuropixels probes. To address this we created a jupyter notebook with step-by-step guidance that is not dependent on the IBL pipeline. This tool (https://github.com/int-brain-lab/paper-reproducible-ephys/blob/develop/RIGOR_script.ipynb) is publicly available through the repository, accompanied by example datasets and usage tutorials.

(2) Table 1: How are qualitative features like "drift" defined? Some quantitative statistics like "presence ratio" (the fraction of the dataset where spikes are present) already exist in packages like ecephys_spike_sorting. Who measured these qualitative features? What are the best practices for doing these qualitative analyses?

At the probe level, we compute the estimate of the relative motion of the electrodes to the brain tissue at multiple depths along the electrode. We overlay the drift estimation over a raster plot to detect sharp displacements as a function of time. Quantitatively, the drift is the cumulative absolute electrode motion estimated during spike sorting (µm). We clarified the corresponding text in Table 1.

The qualitative assessments were carried out by IBL staff and experimentalists. We have now provided code to run the RIGOR metrics along with an embedded tutorial, to complement the supplemental figures we have shown about qualitative metric interpretation.

(3) Table 1: What are the units for the LFP derivative?

We thank the reviewer for noting that the unit was missing. The unit (decibel per unit of space) is now in the table.

(4) Table 1: For "amplitude cutoff", the table says that "each neuron must pass a metric". What is the metric?

We have revised the table to include this information. This metric was designed to detect potential issues in amplitude distributions caused by thresholding during deconvolution, which could result in missed spikes. There are quantitative thresholds on the distribution of the low tail of the amplitude histogram relative to the high tail, and on the relative magnitude of the bins in the low tail. We now reference the methods text from the table, which includes a more extended description and gives the specific threshold numbers. Also, the metric and thresholds are more easily understood with graphical assistance; see the IBL Spike Sorting Whitepaper for this (Fig. 17 in that document and nearby text; https://doi.org/10.6084/m9.figshare.19705522.v4). This reference is now also cited in the text.

(5) Figure 2: In panel A, the brain images look corrupted.

Thanks; in the revised version we have changed the filetype to improve the quality of the panel image.

(6) Figure 7: In panel D, make R2 into R^2 (with a superscript)

Panel D y-axis label has been revised to include superscript (note that this figure is now Figure 8).

Works Cited

Julie M.J. Fabre, Enny H. van Beest, Andrew J. Peters, Matteo Carandini, and Kenneth D. Harris. Bombcell: automated curation and cell classification of spike-sorted electrophysiology data, July 2023. URL https://doi.org/10.5281/zenodo.8172822.

James J. Jun, Nicholas A. Steinmetz, Joshua H. Siegle, Daniel J. Denman, Marius Bauza, Brian Barbarits, Albert K. Lee, Costas A. Anastassiou, Alexandru Andrei, C¸ a˘gatayAydın, Mladen Barbic, Timothy J. Blanche, Vincent Bonin, Jo˜ao Couto, Barundeb Dutta, Sergey L. Gratiy, Diego A. Gutnisky, Michael H¨ausser, Bill Karsh, Peter Ledochowitsch, Carolina Mora Lopez, Catalin Mitelut, Silke Musa, Michael Okun, Marius Pachitariu, Jan Putzeys, P. Dylan Rich, Cyrille Rossant, Wei-lung Sun, Karel Svoboda, Matteo Carandini, Kenneth D. Harris, Christof Koch, John O’Keefe, and Timothy D.Harris. Fully integrated silicon probes for high-density recording of neural activity.Nature, 551(7679):232–236, Nov 2017. ISSN 1476-4687. doi: 10.1038/nature24636. URL https://doi.org/10.1038/nature24636.

Simon Musall, Xiaonan R. Sun, Hemanth Mohan, Xu An, Steven Gluf, Shu-Jing Li, Rhonda Drewes, Emma Cravo, Irene Lenzi, Chaoqun Yin, Bj¨orn M. Kampa, and Anne K. Churchland. Pyramidal cell types drive functionally distinct cortical activity patterns during decision-making. Nature Neuroscience, 26(3):495– 505, Mar 2023. ISSN 1546-1726. doi: 10.1038/s41593-022-01245-9. URL https://doi.org/10.1038/s41593-022-01245-9.

Ivana Orsolic, Maxime Rio, Thomas D Mrsic-Flogel, and Petr Znamenskiy. Mesoscale cortical dynamics reflect the interaction of sensory evidence and temporal expectation during perceptual decision-making. Neuron, 109(11):1861–1875.e10, April 2021. Hyeong-Dong Park, St´ephanie Correia, Antoine Ducorps, and Catherine Tallon-Baudry.Spontaneous fluctuations in neural responses to heartbeats predict visual detection.Nature Neuroscience, 17(4):612–618, Apr 2014. ISSN 1546-1726. doi: 10.1038/nn.3671. URL https://doi.org/10.1038/nn.3671.

Lorenzo Posani, Shuqi Wang, Samuel Muscinelli, Liam Paninski, and Stefano Fusi. Rarely categorical, always high-dimensional: how the neural code changes along the cortical hierarchy. bioRxiv, 2024. doi: 10.1101/2024.11.15.623878. URL https://www.biorxiv.org/content/early/2024/12/09/2024.11.15.623878.

Nicholas A. Steinmetz, Christina Buetfering, Jerome Lecoq, Christian R. Lee, Andrew J. Peters, Elina A. K. Jacobs, Philip Coen, Douglas R. Ollerenshaw, Matthew T. Valley, Saskia E. J. de Vries, Marina Garrett, Jun Zhuang, Peter A. Groblewski, Sahar Manavi, Jesse Miles, Casey White, Eric Lee, Fiona Griffin, Joshua D. Larkin, Kate Roll, Sissy Cross, Thuyanh V. Nguyen, Rachael Larsen, Julie Pendergraft, Tanya Daigle, Bosiljka Tasic, Carol L. Thompson, Jack Waters, Shawn Olsen, David J. Margolis, Hongkui Zeng, Michael Hausser, Matteo Carandini, and Kenneth D. Harris. Aberrant cortical activity in multiple gcamp6-expressing transgenic mouse lines. eNeuro, 4(5), 2017. doi: 10.1523/ENEURO.0207-17.2017. URL https://www.eneuro.org/content/4/5/ENEURO.0207-17.2017.

Nicholas A. Steinmetz, Peter Zatka-Haas, Matteo Carandini, and Kenneth D. Harris. Distributed coding of choice, action and engagement across the mouse brain. Nature, 576(7786):266–273, Dec 2019. ISSN 1476-4687. doi: 10.1038/s41586-019-1787-x. URL https://doi.org/10.1038/s41586-019-1787-x.

Nicholas A. Steinmetz, Cagatay Aydin, Anna Lebedeva, Michael Okun, Marius Pachitariu, Marius Bauza, Maxime Beau, Jai Bhagat, Claudia B¨ohm, Martijn Broux, Susu Chen, Jennifer Colonell, Richard J. Gardner, Bill Karsh, Fabian Kloosterman, Dimitar Kostadinov, Carolina Mora-Lopez, John O’Callaghan, Junchol Park, Jan Putzeys, Britton Sauerbrei, Rik J. J. van Daal, Abraham Z. Vollan, Shiwei Wang, Marleen Welkenhuysen, Zhiwen Ye, Joshua T. Dudman, Barundeb Dutta, Adam W. Hantman,Kenneth D. Harris, Albert K. Lee, Edvard I. Moser, John O’Keefe, Alfonso Renart, Karel Svoboda, Michael H¨ausser, Sebastian Haesler, Matteo Carandini, and Timothy D. Harris. Neuropixels 2.0: A miniaturized high-density probe for stable, long-term brain recordings. Science, 372(6539):eabf4588, 2021. doi: 10.1126/science.abf4588.URL https://www.science.org/doi/abs/10.1126/science.abf4588.

Charlie Windolf, Han Yu, Angelique C. Paulk, Domokos Mesz´ena, William Mu˜noz, Julien Boussard, Richard Hardstone, Irene Caprara, Mohsen Jamali, Yoav Kfir, Duo Xu, Jason E. Chung, Kristin K. Sellers, Zhiwen Ye, Jordan Shaker, Anna Lebedeva, Manu Raghavan, Eric Trautmann, Max Melin, Jo˜ao Couto, Samuel Garcia, Brian Coughlin, Csaba Horv´ath, Rich´ard Fi´ath, Istv´an Ulbert, J. Anthony Movshon, Michael N. Shadlen, Mark M. Churchland, Anne K. Churchland, Nicholas A. Steinmetz, Edward F. Chang, Jeffrey S. Schweitzer, Ziv M. Williams, Sydney S. Cash, Liam Paninski, and Erdem Varol. Dredge: robust motion correction for high-density extracellular recordings across species. bioRxiv, 2023. doi: 10.1101/2023.10.24.563768. URL https://www.biorxiv.org/content/early/2023/10/29/2023.10.24.563768.